# Robust Learning of Optimal Auctions

**Wenshuo Guo**
University of California, Berkeley
wguo@cs.berkeley.edu

**Michael I. Jordan**
University of California, Berkeley
jordan@cs.berkeley.edu

**Manolis Zampetakis**
University of California, Berkeley
mzampet@berkeley.edu

## Abstract

We study the problem of learning revenue-optimal multi-bidder auctions from samples when the samples of bidders' valuations can be adversarially corrupted or drawn from distributions that are adversarially perturbed. First, we prove tight upper bounds on the revenue we can obtain with a corrupted distribution under a population model, for both regular valuation distributions and distributions with monotone hazard rate (MHR). We then propose new algorithms that, given only an "approximate distribution" for the bidder's valuation, can learn a mechanism whose revenue is nearly optimal simultaneously for all "true distributions" that are $\alpha$-close to the original distribution in Kolmogorov-Smirnov distance. The proposed algorithms operate beyond the setting of bounded distributions that have been studied in prior works, and are guaranteed to obtain a fraction $1 - O(\alpha)$ of the optimal revenue under the true distribution when the distributions are MHR. Moreover, they are guaranteed to yield at least a fraction $1 - O(\sqrt{\alpha})$ of the optimal revenue when the distributions are regular. We prove that these upper bounds cannot be further improved, by providing matching lower bounds. Lastly, we derive sample complexity upper bounds for learning a near-optimal auction for both MHR and regular distributions.

## 1 Introduction

Optimal auctions play a crucial role in economic theory, with a wide range of applications across various industries, public sectors, and online platforms [Myerson, 1981, Bykowsky et al., 2000, Roth and Ockenfels, 2002, Klemperer, 2002, Milgrom and Milgrom, 2004, Lahaie et al., 2007]. In such auctions, pricing mechanisms need to be determined by the auction designer so as to satisfy various desired goals, such as revenue maximization and incentive compatibility. Often this determination is made based on information about the buyers that is assumed to be available a priori. For example, in a standard valuation model, each bidder has a valuation over the available items, and if the sellers knows the distribution of these valuations, they could design an optimal auction which maximizes the revenue.

Arguably the fundamental difficulty in the design of optimal auctions is that real valuations are private and unknown to the auction designer. Consider specifically the problem of selling one item to multiple buyers. Suppose that we model the buyers' valuations as arising as independent draws from buyer-specific prior distributions. In this scenario, what is the optimal mechanism in terms of the expected revenue? This problem was solved by Myerson [1981] through a characterization of *virtual value functions*. In particular, we can define a virtual value function of each buyer based on their

prior distributions. An optimal auction then lets the buyer with the largest non-negative virtual value win the item, and charges the winner a price that equals the threshold value above which she wins.[1]

Unfortunately, there is a further fundamental challenge in deploying these theoretical results in practice, which is that in real-world settings the auction designer may not even know the prior distributions on valuations. Instead, what the designer might hope for is that there is a stream of previous transactions, or some other relevant auxiliary data, that is helpful in inferring the buyers' private distributions. This perspective has motivated an active recent literature learning optimal auctions from samples [Cole and Roughgarden, 2014, Devanur et al., 2016, Morgenstern and Roughgarden, 2015, 2016, Syrgkanis, 2017, Dudík et al., 2017, Gonczarowski and Nisan, 2017, Huang et al., 2018, Roughgarden and Schrijvers, 2016, Balcan et al., 2018, Guo et al., 2019, Roughgarden and Wang, 2019, Gonczarowski and Weinberg, 2021]. In this line of work, the central question is: suppose we are only able to access the prior distributions in the form of independent samples, how many samples are sufficient and necessary for finding an approximately optimal auction?

While this merging of mechanism design and learning theory is appealing, a further concern arises. Given the potentially adversarial setting of auction design, do we really believe that the data that we observe are drawn in accord with our assumptions? More concretely, is the learning of optimal auctions robust to adversarial corruptions of the samples? This problem is arguably at the core of what it means to learn an optimal auction. It is a challenging problem; indeed, as we show in Counterexample 1 in Section 4, auction designs that are optimal in the absence of corruptions can become arbitrarily bad even if a small portion of the samples are corrupted. Building on earlier work by Cai and Daskalakis [2017] and Brustle et al. [2020], we tackle a key open problem—what is the best approximation to the optimal revenue for arbitrary levels of corruption for distributions with unbounded support? And what is the mechanism that achieves it?

In summary, in this work we explore the problem of the robust learning of optimal auctions, where the samples of bidders' valuations are subject to corruption and their support is unbounded. In particular, we consider having access to samples that are drawn from some distribution $\tilde{\mathcal{D}}$ which is within a Kolmogorov-Smirnov (KS) distance $\alpha$ of the true distribution $\mathcal{D}^*$. Denote OPT as the maximum revenue we can achieve under the true valuation distributions. Our goal is to design mechanisms that are guaranteed to achieve a revenue of at least $(1 - \rho(\alpha)) \cdot \text{OPT}$ for the smallest possible error $\rho(\alpha)$ and with the use of a minimal number of samples.

## 1.1 Our results

We study the problem of learning revenue-optimal multi-bidder auctions from samples when the samples of bidders' valuations can be adversarially corrupted or drawn from distributions that are adversarially perturbed. We summarize our main results as follows:

1. We derive tight upper bounds on the revenue we can obtain with a corrupted distribution under a population model. For distributions with monotone hazard rate (MHR), and with total corruption $\alpha$, we obtain an approximation ratio of $1 - O(\alpha)$ compared to the optimal revenue under the true distribution (see Theorem 3.6). For regular valuation distributions, where for total corruption $\alpha$, we get an approximation ratio of $1 - O(\sqrt{\alpha})$ (see Theorem 3.8).

2. To achieve these upper bounds, we propose a new *theoretical* algorithm for the population model (see Algorithm 1) that, given only an "approximate distribution" for the bidder's valuation, can learn a mechanism whose revenue is nearly optimal simultaneously for all "true distributions" that are $\alpha$-close to the given distribution in Kolmogorov-Smirnov distance. The proposed algorithm operates beyond the setting of bounded distributions that have been studied in prior works; indeed, they apply to general unbounded MHR and regular distributions.

3. We further show that these upper bounds under the population model cannot be further improved (up to constant log factors), by providing matching lower bounds for both the MHR and regular distributions (see Theorem 3.7 and Theorem 3.9).

4. Lastly, we derive sample complexity upper bounds for learning a near-optimal auction for both MHR and regular distributions with multiple bidders (Theorem 4.3 and Theorem 4.4), and propose a *practical* algorithm (see Algorithm 2) which takes samples as input. We also

---

[1]More generally, the optimal auction picks the winner based on the virtual value after an "ironing" procedure.

provide accompanying sample complexity lower bounds (Theorem 4.5), and demonstrate a small gap relative to the corresponding upper bounds.

## 1.2 Related work

Designing revenue optimal auctions is a classic problem in economic theory that has attracted much research attention. We survey the most closely related work in two main areas.

**Learning optimal auctions from samples.** Recent work has explored settings of learning approximately optimal auction from samples, both for single-item auctions [Cole and Roughgarden, 2014], and multi-item auctions [Balcan et al., 2018, 2016, Morgenstern and Roughgarden, 2015, Syrgkanis, 2017]. Most recently, Guo et al. [2019] provide a complete set of sample complexity bounds for single-item auctions, by deriving matching upper and lower bounds up to a poly-logarithmic factor. While these approaches have obtained fruitful results on the sample complexity of learning optimal auctions, a key assumption that is commonly made in this work is that the samples are independently and identically drawn from the bidders' valuation distributions, with the goal of learning an auction which maximizes the expected revenue on the underlying, unknown distribution over bidder valuations. A major difference in our work is that we consider that the samples can suffer from potential corruptions, which is a significantly more challenging setting.

**Robustness of learning optimal auctions.** Our paradigm on the robust learning of optimal auctions is closely related to recent work that considers the learning of auctions from mismatched distributions or corrupted samples. Cai and Daskalakis [2017] consider a multi-item auction setting, where there is a given "approximate distribution," and the goal is to compute an auction whose revenue is approximately optimal simultaneously for all "true distributions" that are close to the given one. They provide an algorithm that achieves a poly-$\alpha$ additive loss compared to the true optimal revenue. More recently, Brustle et al. [2020] consider learning multi-item auctions where bidders' valuations are drawn from correlated distributions that can be captured by Markov random fields. However, they make a key simplifying assumption—that the bidders' valuation for the items lie in some bounded interval. Our results, by contrast, apply to the general setting of unbounded valuation distributions, a setting that requires new theoretical machinery. To the best of our knowledge, our work constitutes the first analysis of the learnability of single-item optimal auctions from corrupted samples for unbounded distributions.

**Organization.** In Section 2, we provide background on auction models and formally state our problem. Section 3 contains our main theoretical statements for the population model. We propose an algorithm that achieves optimal theoretical upper bounds, by providing matching lower bounds. Section 4 contains our main results on learning with finite samples. We provide a practical algorithm that takes samples from the corrupted distribution, and provides sample complexity upper and lower bounds for both the regular and MHR distributions cases. We conclude in Section 5.

## 2 Preliminaries

We begin by formally defining the setting we study for robust learning of optimal auctions, which includes the revenue objective and the general classes of valuation distributions that we consider.

### 2.1 Auction models

**Single-bidder setting.** Consider one item for sale to one bidder. The bidder has a private valuation $v \in \mathbb{R}_+$ for this item. We assume that $v$ is a random variable distributed according to the distribution $\mathcal{D}^*$, with support $\mathbb{R}_+$, cumulative distribution function $F$, and probability density function $f$.

It is well known that the optimal auction in this setting is a reserve price auction, such that the task for the seller is to compute a reserve price $p$ that optimizes revenue [Myerson, 1981]. We assume that the bidder has a quasi-linear utility that is equal to $u(p) = v - p$ if she decides to buy the item and $u(p) = 0$ otherwise. The seller aims to set $p$ such that her expected revenue—i.e., the received payment—is maximized. We consider the setting where both $v$ and $\mathcal{D}^*$ are unknown to the seller.

However, the seller can access i.i.d. samples that are drawn from a distribution $\tilde{\mathcal{D}}$, which is $\alpha$-close to $\mathcal{D}$ with regard to the Kolmogorov distance:

**Definition 2.1.** (Kolmogorov-Smirnov distance) For probability measures $\mu$ and $\nu$ on $\mathbb{R}$, define

$$d_k(\mu, \nu) = \sup_{x \in \mathbb{R}} |\mu((-\infty, x)) - \nu((-\infty, x))|.$$

It is well known that $d_k(\mu, \nu) \leqslant d_{TV}(\mu, \nu)$, where $d_{TV}$ denotes the total variation (TV) distance between $\mu$ and $\nu$. The closeness of $\tilde{\mathcal{D}}$ to $\mathcal{D}^*$ is thus formalized as follows:

$$d_k(\mathcal{D}^*, \tilde{\mathcal{D}}) \leqslant \alpha,$$

for some $\alpha > 0$.

**Multi-bidder setting.** Consider one item for sale to $n$ bidders. Each bidder has a private valuation, $v_i \in \mathbb{R}_+$, where $v_i$ is independently drawn from the corresponding prior distribution $\mathcal{D}_i^*$. Thus, the valuations $\mathbf{v} = (v_1, v_2, \cdots, v_n)$ follow a product distribution $\mathbf{D}^* = \mathcal{D}_1^* \times \cdots \times \mathcal{D}_n^*$. Each bidder submits a bid $b_i \geqslant 0$. Denote all the bids as $\mathbf{b} = (b_1, \cdots, b_n)$. A mechanism in this setting consists of two rules: the allocation rule $\mathbf{x}(\mathbf{b})$ that takes the bids $\mathbf{b}$ and outputs the probability $x_i(\mathbf{b})$ that each bidder $i$ will receive the item, and the payment rule $\mathbf{p}(\mathbf{b})$ that takes the bids $\mathbf{b}$ and outputs the payment of bidder $i$. Bidder $i$'s utility is then $u_i(\mathbf{b}) = v_i \cdot x_i(\mathbf{b}) - p_i(\mathbf{b})$. The goal of the seller is to find a mechanism that maximizes the expected revenue $\mathbb{E}[\sum_{i \in [n]} p_i(\mathbf{b})]$, where the expectation is over $\mathbf{v} \sim \mathbf{D}^*$, under the following *Dominant Strategy Incentive Compatibility (DSIC)* and the *Individual Rationality (IR)* constraints:

$$u_i(v_i, \mathbf{b}_{-i}) \geqslant u_i(b_i, \mathbf{b}_{-i}) \qquad \text{for all } v_i, b_i \in \mathbb{R}_+ \text{ and all } \mathbf{b}_{-i} \in \mathbb{R}_+^{n-1} \qquad \text{(DSIC)}$$

$$u_i(v_i, \mathbf{b}_{-i}) \geqslant 0 \qquad \text{for all } v_i \in \mathbb{R}_+ \text{ and all } \mathbf{b}_{-i} \in \mathbb{R}_+^{n-1}. \qquad \text{(IR)}$$

We consider the setting in which the valuations and the prior distributions are unknown to the seller. Instead, the seller has access to a finite number of i.i.d. samples drawn from the product distribution $\tilde{\mathcal{D}} = \tilde{\mathcal{D}}_1 \times \cdots \times \tilde{\mathcal{D}}_n$, where each $\tilde{\mathcal{D}}_i$ satisfies

$$d_k(\mathcal{D}_i^*, \tilde{\mathcal{D}}_i) \leqslant \alpha_i,$$

for some $\alpha_i > 0, \forall i \in [n]$.

**Revenue objective.** Letting $\mathbf{D}$, $\mathbf{D}'$ be product or single bidder distributions as described above, we define $M_{\mathbf{D}}$ to be the mechanism that achieves the optimal revenue for the value distributions $\mathbf{D}$ and $\text{OPT}(\mathbf{D})$ its expected revenue. Let also $\text{Rev}(M_{\mathbf{D}}, \mathbf{D}')$ be the expected revenue of the mechanism $M_{\mathbf{D}}$ when applied to a setting where the values are drawn with respect to $\mathbf{D}'$.

## 2.2 Monotone hazard rate (MHR) and regular distributions

For any bidder $i$ with a valuation $v_i \sim \mathcal{D}_i$, define the *virtual value function* for this bidder as $\phi_i(v) \stackrel{\text{def}}{=} v - \frac{1 - F_i(v)}{f_i(v)}$, where $F_i$ and $f_i$ are the CDF and PDF of $\mathcal{D}_i$. The *hazard rate* of the distribution $\mathcal{D}_i$ is defined as the function $\frac{f_i(v)}{1 - F_i(v)}$. Then, the distribution $\mathcal{D}_i$ is said to be *regular* if the virtual value $\phi_i(v)$ is monotonically non-decreasing in $v$. Further, distribution $\mathcal{D}_i$ has *monotone hazard rate* (MHR) if $\frac{f_i(v)}{1 - F_i(v)}$ is monotone non-decreasing.

## 3 The Population Model

In this section, we study the problem of learning optimal auction assuming that we have the exact knowledge of the adversarially perturbed distributions $\tilde{\mathbf{D}}$. We relax this assumption in Section 4 where we show how to learn optimal auctions when we only have sample access to $\tilde{\mathbf{D}}$.

We begin in Section 3.1 with the description of our mechanism in the population model. Then, in Section 3.2, we present our analysis for the population mechanism for Monotone Hazard Rate distributions and we also present the sketch of our proof for the single-bidder case. Similarly, in Section 3.3 we state our analysis for the population mechanism for regular distributions and we present a proof sketch for the single-bidder case. Finally, we show that our proposed mechanism achieves optimal (up to constants) guarantees among any mechanism in the population model.

## 3.1 Robust Myerson auction in the population model

Our algorithm assumes as an input the exact knowledge of a product distribution, $\tilde{\mathbf{D}} = \tilde{\mathcal{D}}_1 \times \cdots \times \tilde{\mathcal{D}}_n$, such that the $d_k(\mathcal{D}_i^*, \tilde{\mathcal{D}}_i) \leqslant \alpha_i$ and its goal is to find a mechanism that achieves approximately optimal revenue for $\mathbf{D}^*$, where $\mathbf{D}^* = \Pi_i \mathcal{D}_i^*$. Without further assumptions, this is an impossible task, as we explain in Section 4 via an example. Thus we assume that the algorithm possesses some additional knowledge regarding $\mathcal{D}_i^*$, either that it is MHR or regular, and the mechanism needs to exploit this additional property.

To utilize the additional property of the distributions $\mathcal{D}_i^*$, our mechanism uses the important concept of the *link function* for MHR and regular distributions.

**Definition 3.1** (Link Function). The link function $h_M(x; F)$ for MHR distributions is defined as $h_M(x; F) = -\ln(1 - F(x))$ and the link function $h_r(x; F)$ for regular distributions is defined as $h_r(x; F) = 1/(1 - F(x))$. We also define the corresponding inverse link functions $h_M^{-1}(x; h) = 1 - \exp(-h(x))$ and $h_r^{-1}(x; h) = 1 - 1/h(x)$. Observe that $h_M^{-1}(x; h_M(x; F)) = F(x)$ and $h_r^{-1}(x; h_r(x; F)) = F(x)$. We may write $h_M(x)$ or $h_r(x)$ when $F$ is clear from the context.

We provide some intuition on the link function. First, by construction, the link function of either an MHR distribution or a regular distribution is convex and non-decreasing. Second, the link function is monotone with regard to $F$. These two properties are important when we define the notion of a minimal MHR/regular distribution in a Kolmogorov ball, momentarily, which will be used as a necessary step in our algorithm.

Importantly, the link function provides a convenient characterization of the optimal reserve price and optimal revenue for a distribution $F$ that is MHR or regular. To see this, first consider a single bidder with a valuation distribution $F$. Denote the optimal reserve price for selling one item to her as $x^*$, and the optimal expected revenue as $\mathrm{OPT}(F)$. Then, when $F$ is MHR, we show that $x^*$ is also the unique minimizer of $(h_M(x) - \log(x))$. On the other hand, when $F$ is regular, $v^*$ is the point where $h_r(x)$ intersects with its tangent line $kx$, with $k = 1/\mathrm{OPT}(F)$ (proof details in Appendix). Figure 1 illustrates such a useful property for $h_M$ and $h_r$ explicitly, for a single-item, single-bidder auction.

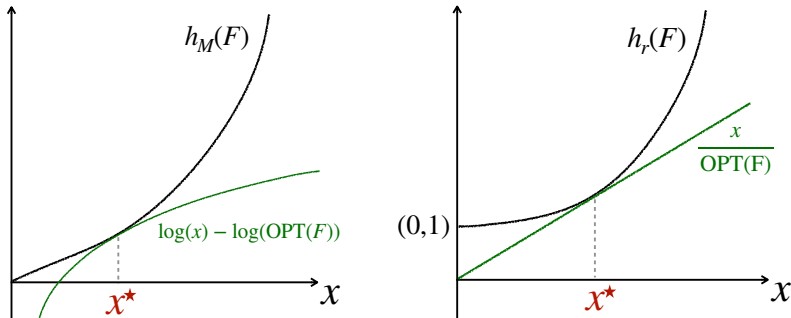

Figure 1: Optimal reserve price $x^*$ with regard to the link function, for a single-item single-bidder auction with a valuation distribution $F$. *(left)* $F$ is MHR; *(right)* $F$ is regular.

Next, we formally define stochastic dominance between two distributions, and state the property of strong revenue monotonicity.

**Definition 3.2** (Stochastic dominance). Given two distributions $\mathcal{D}_1$ and $\mathcal{D}_2$ with CDFs as $F_1$ and $F_2$. Then, we say $\mathcal{D}_1$ (first-order) stochastically dominates $\mathcal{D}_2$ if for every $x \in \mathcal{X}$,

$$F_1(x) \leqslant F_2(x),$$

denoted as $\mathcal{D}_1 \succeq \mathcal{D}_2$. We say a product distribution $\mathbf{D} = \Pi_i \mathcal{D}_i$ (component-wise) stochastically dominates another product distribution $\mathbf{D}' = \Pi_i \mathcal{D}_i'$ if for every $i$, we have $\mathcal{D}_i \succeq \mathcal{D}_i'$.

**Lemma 3.3** (Strong revenue monotonicity [Guo et al., 2019]). *Let $\mathbf{D}$, $\mathbf{D}'$ be two product distributions such that $\mathbf{D}' \succeq \mathbf{D}$, then, for $M$ that is the optimal mechanism for $\mathbf{D}$, we have:*

$$\mathrm{Rev}(M, \mathbf{D}) \leqslant \mathrm{Rev}(M, \mathbf{D}').$$

**Algorithm 1** Robust Myerson Auction in the Population Model

1: **Input:** $\alpha_1 \dots \alpha_n > 0$, link function $h(\cdot)$, possibly corrupted valuation distribution $\tilde{F} = \Pi_{i=1}^n \tilde{F}_i$.
2: **for** i = 1 … n **do**
3: Compute a minimal regular / MHR distribution in $B_{d_k, \alpha_i}(\tilde{F}_i)$ according to Eq (1), denote as $\widehat{F}_i$.
4: **end for**
5: Set $\widehat{F} = \Pi_{i=1}^n \widehat{F}_i$.
6: Output Myerson's optimal auction $M_{\widehat{F}}$ w.r.t. the distribution $\widehat{F}$.

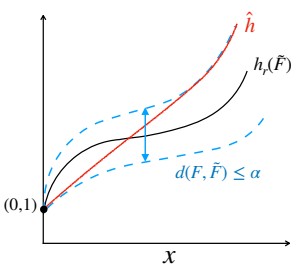

Figure 2: A minimal regular distribution in $B_{d_k, \alpha}$, in the space transformed by applying the link function.

The following lemma illustrates the importance of the link functions as well as their connection with first-order stochastic dominance. The proof of this lemma is given in Appendix A.

**Lemma 3.4.** *A distribution with CDF $F$ is MHR if and only if $h_M(x; F)$ is a convex function of $x$. Similarly, $F$ is regular if and only if $h_r(x; F)$ is a convex function of $x$. Moreover, for two MHR (resp. regular) distributions $F_1$ and $F_2$, such that $F_1 \succeq F_2$, we have that $h_M(x; F_1) \leqslant h_M(x; F_2)$ (resp. $h_r(x; F_1) \leqslant h_r(x; F_2)$) for all $x$.*

A key idea used in our algorithm is the minimal MHR/regular distribution within a Kolmogorov distance divergence ball. Formally,

**Definition 3.5.** For a given distribution with its cumulative distribution function as $F$, denote the set of all the distributions that are $\alpha$-close to $F$ in Kolmogorov distance as $B_{d_k, \alpha}(F)$:

$$B_{d_k, \alpha}(F) \overset{\text{def}}{=} \{F' : d_k(F', F) \leqslant \alpha\}.$$

Further, define a minimal MHR/regular distribution within $B_{d_k, \alpha}(F)$ as:

$$\widehat{F}(x) = h^{-1}(x; \widehat{h}), \quad \text{where} \quad \widehat{h}(x) \overset{\text{def}}{=} \max_{\substack{\tilde{F} \in B_{d_k, \alpha}(F) \\ \tilde{F} \text{ is MHR / regular}}} h\left(\tilde{F}(x)\right) \quad \forall x \in \mathbb{R}_+. \tag{1}$$

Figure 2 gives an illustration of a minimal regular distribution within $B_{d_k, \alpha}(F)$, in the space transformed by the link function of regular distributions.

### 3.2 Analysis for MHR distributions

In this section we state the results for the performance of Algorithm 1 for MHR distributions and we provide a proof sketch for the single-bidder case. The full proof of the following theorem can be found in Appendix B.

**Theorem 3.6.** *Let $\mathbf{D}^* = \mathcal{D}_1^* \times \cdots \times \mathcal{D}_n^*$ be a product distribution where every $\mathcal{D}_i^*$ is MHR. Let also $\tilde{\mathbf{D}} = \tilde{\mathcal{D}}_1 \times \cdots \tilde{\mathcal{D}}_n$ be any product distribution such that for all $i \in [n]$ it holds that $d_k(\mathcal{D}_i^*, \tilde{\mathcal{D}}_i) \leqslant \alpha_i$. If $\tilde{M}$ is the mechanism that Algorithm 1 outputs with input $\tilde{\mathbf{D}}$ then it holds that*

$$\text{Rev}(\tilde{M}, \mathbf{D}^*) \geqslant \left(1 - \tilde{O}\left(\sum_{i=1}^n \alpha_i\right)\right) \cdot \text{OPT}(\mathbf{D}^*).$$

*In particular for $n = 1$, if $\alpha = \alpha_1$, then we have that $\text{Rev}(\tilde{M}, \mathcal{D}^*) \geqslant (1 - O(\alpha)) \cdot \text{OPT}(\mathcal{D}^*)$.*

*Proof sketch for $n = 1$.* The first key step in our proof is the observation that, by construction, Algorithm 1 runs the Myerson optimal auction on an MHR distribution $\widehat{F}$, such that $\widehat{F}$ is stochastically dominated by any other MHR distribution that is within $B_{d_k, \alpha}(\tilde{F}')$. On the other hand we have $d_k(F^*(x), \tilde{F}(x)) \leqslant \alpha$. Applying the triangle inequality, we have $d_k(F^*(x), \widehat{F}(x)) \leqslant 2\alpha$. It is then sufficient for us to bound the ratio of the optimal revenue for any two MHR distributions $F_1$ and $F_2$, with $d_k(F_1, F_2) \leqslant 2\alpha$, and where $F_1$ is stochastically dominated by $F_2$.

The key part of our proof then considers such $F_1$, $F_2$, and due to the fact that the ratio of the revenues, $\text{OPT}_{F_1}/\text{OPT}_{F_2}$, is scale invariant, we assume without loss of generality that $\text{OPT}_{F_1} = 1$. We then prove that this leads to $h(P^*_{F_1}) \leqslant 1$. The result then follows from two further key lemmas. First, for any reserve price $x < P^*_{F_1}$, $|h_1(x) - h_2(x)| = \left|\log\left(\frac{1-F_2(x)}{1-F_1(x)}\right)\right|$. Further applying the fact that by assumption $|F_1(x) - F_2(x)| \leqslant \alpha$ we show that $|h_1(x) - h_2(x)| = O(\alpha)$ for any reserve price $x < P^*_{F_1}$. Second, using the fact that $F_1$ is stochastically dominated by $F_2$, we derive that $P^*_{F_2} \leqslant P^*_{F_1}$. The conclusion then follows from bounding the ratio of $s_1(x) = h_1(x) - \log(x)$, and $s_2(x) = h_2(x) - \log(x)$, based on the definition of $P^*_{F_1}$ and $P^*_{F_2}$. ∎

Next we show that the information-theoretic Algorithm 1 is optimal up to constants for MHR distributions. We provide the proof of the following theorem in Appendix C.

**Theorem 3.7.** *Let $M$ be any DSIC and IR mechanism that takes as input a product distribution $\tilde{\mathbf{D}} = \hat{\mathcal{D}}_1 \times \cdots \times \hat{\mathcal{D}}_n$. Then there exists a product distribution $\mathbf{D}^* = \mathcal{D}_1^* \times \cdots \times \mathcal{D}_n^*$ such that $d_k(\mathcal{D}_i^*, \tilde{\mathcal{D}}_i) \leqslant \alpha$, $\mathcal{D}_i^*$ is MHR for every $i$, and*

$$\text{Rev}(M, \mathbf{D}^*) \leqslant (1 - \tilde{\Omega}(n \cdot \alpha)) \cdot \text{OPT}(\mathbf{D}^*).$$

### 3.3 Analysis for regular distributions

In this section we state the results for the performance of Algorithm 1 for regular distributions and we provide a proof sketch for the single-bidder case. The full proof of the following theorem can be found in Appendix B.

**Theorem 3.8.** *Let $\mathbf{D}^* = \mathcal{D}_1^* \times \cdots \times \mathcal{D}_n^*$ be a product distribution where every $\mathcal{D}_i^*$ is regular. Let also $\tilde{\mathbf{D}} = \tilde{\mathcal{D}}_1 \times \cdots \tilde{\mathcal{D}}_n$ be any product distribution such that for all $i \in [n]$ it holds that $d_k(\mathcal{D}_i^*, \tilde{\mathcal{D}}_i) \leqslant \alpha_i$. If $\tilde{M}$ is the mechanism that Algorithm 1 outputs with input $\tilde{\mathbf{D}}$ then it holds that*

$$\text{Rev}(\tilde{M}, \mathbf{D}^*) \geqslant \left(1 - 5 \cdot \sqrt{\sum_{i=1}^{n} \alpha_i}\right) \cdot \text{OPT}(\mathbf{D}^*).$$

*Proof sketch for $n = 1$.* We first prove a general result that for two regular distributions $F$ and $\bar{F}$, such that $d_k(F, \bar{F}) \leqslant \alpha$, where $F(x)$ is stochastically dominated by $\bar{F}(x)$ for $x \in \mathbb{R}_+$. The optimal revenue of these two distributions is close, formally $\frac{\text{OPT}(F)}{\text{OPT}(\bar{F})} \geqslant 1 - O(\sqrt{\alpha})$. The first key step replies on using the link function $h_r(x) = \frac{1}{1-F(x)}$ for regular distributions. Since $h_r(x)$ preserves the same monotonicity property as $F(x)$, we first derive a lower bound on $\bar{h}_r(x, \bar{F})$ that is $\bar{h}_r(x, \bar{F}) \geqslant h_r(x, F) - \alpha h_r^2(x, F)$, using the fact that $d_k(F, \bar{F}) \leqslant \alpha$. This bound gives us useful constraints to discuss in different cases in the following part of the proof. Denote the corresponding optimal reserve prices for $F$ and $\bar{F}$ as $P$ and $\bar{P}$. We discuss separately two cases for $h(\bar{P})$, where, for case 1 we have $h(\bar{P}) \leqslant \frac{1}{\sqrt{\alpha}}$, and for case 2, we have $h(\bar{P}) > \frac{1}{\sqrt{\alpha}}$. Using the connection from the link function to the revenue (see Figure 1), case 1 directly leads to the conclusion that $\frac{\text{OPT}(F)}{\text{OPT}(\bar{F})} \geqslant 1 - \sqrt{\alpha}$. Case 2 is more subtle and requires a more careful argument. Lastly, by construction, Algorithm 1 runs the Myerson optimal auction on a regular distribution $\widehat{F}$, such that $\widehat{F} \geqslant \widehat{F}'(x)$ for all $x \in \mathbb{R}_+$, for any other regular distribution $F'(x)$ such that $d_k(F'(x), \tilde{F}(x)) \leqslant \alpha$. Applying the triangle inequality and combining with the conclusions obtained from the two cases concludes the proof. ∎

Finally, we show that the information-theoretic Algorithm 1 is optimal up to constants for regular distributions. We provide the proof of the following theorem in Appendix C.

**Theorem 3.9.** *Let $M$ be any DSIC and IR mechanism that takes as input a product distribution $\tilde{\mathbf{D}} = \tilde{\mathcal{D}}_1 \times \cdots \times \tilde{\mathcal{D}}_n$. Then there exists a product distribution $\mathbf{D}^* = \mathcal{D}_1^* \times \cdots \times \mathcal{D}_n^*$ such that $d_k(\mathcal{D}_i^*, \tilde{\mathcal{D}}_i) \leqslant \alpha$, $\mathcal{D}_i^*$ is regular for every $i$, and*

$$\text{Rev}(M, \mathbf{D}^*) \leqslant (1 - \Omega(\sqrt{n \cdot \alpha})) \cdot \text{OPT}(\mathbf{D}^*).$$

# 4 Finite Samples

We provide a practical algorithm that takes samples from the corrupted distribution $\tilde{\mathbf{D}}$ as an input. We show that this algorithm achieves almost optimal sample complexity for the MHR distribution case and the single-bidder regular distribution case, whereas for the multi-bidder regular distributions there is a small gap between our upper and lower bounds.

An important notion to explain our algorithm for the finite-sample case is the following notion of the convex envelope.

**Definition 4.1** (Convex Envelope). The convex envelope $Conv(f)$ of a function $f$ is a function with the following property

$$Conv(f)(x) = \sup\{g(x) \mid g \text{ is convex and } g \leqslant f \text{ over } \mathbb{R}_+\}.$$

In words, $Conv(f)$ is the maximum convex function that is below $f$.

For our algorithm one important property of the convex envelope is expressed in the following lemma whose proof is presented in Appendix A.

**Lemma 4.2.** *Let $f$ be a non-decreasing piecewise constant function with $k$ pieces, then $Conv(f)$ can be computed in time $\mathrm{poly}(k)$ and is a piecewise linear function with $O(k)$ pieces.*

---

**Algorithm 2** Robust Empirical Myerson Auction

1: **Input:** $m$ i.i.d. samples from (possibly corrupted) value distribution $\mathbf{D} = \Pi_{i=1}^n \mathcal{D}_i$, link function $h(\cdot)$.
2: Let $\mathbf{E} = \Pi_{i=1}^n E_i$ be the empirical distribution, i.e., the uniform distribution over the samples.
3: **for** $i = 1 \dots n$ **do**
4:     Construct $\widehat{E_i}$ as following: let $q^{E_i}(v)$ be the quantile of $E_i$; the quantile of $\widehat{E_i}$ is as follows:

$$q^{\widehat{E_i}}(v) = \begin{cases} \max\left\{0, q^{E_i}(v) - \sqrt{\frac{2q^{E_i}(v)\left(1 - q^{E_i}(v)\right)\ln(2mn\delta^{-1})}{m}} - \frac{4\ln(2mn\delta^{-1})}{m} - \alpha_i\right\} & \text{if } v > 0 \\ 1 & \text{if } v = 0 \end{cases}$$

5:     Construct $\tilde{E}_i$ such that $h\left(\tilde{E}_i(\cdot)\right)$ is the convex envelope of $h\left(\widehat{E}(\cdot)\right)$, i.e.

$$\tilde{E}_i(\cdot) = h^{-1}\left(Conv\left(h(\widehat{E}_i(\cdot))\right)\right)$$

6: **end for**
7: Set $\tilde{\mathbf{E}} = \Pi_{i=1}^n \tilde{E}_i$
8: Output Myerson's optimal auction $M_{\tilde{\mathbf{E}}}$ w.r.t. $\tilde{\mathbf{E}}$.

---

The above algorithm resembles the main algorithm of Guo et al. [2019] with the addition of step 5. We first show that step 5 is necessary if we wish to obtain any non-trivial result in the robust auction learning setting that we explore in this paper.

**Counterexample 1.** Imagine we have just one agent, i.e., $n = 1$, with true distribution $\mathcal{D}^*$ equal to an exponential distribution with parameter $\lambda = 1$. Also, to strengthen our counterexample imagine that we have available an infinite number of samples, i.e., $m \to \infty$. Now consider $\tilde{\mathcal{D}}$ to be the corrupted distribution where probability mass $\alpha$ is removed from the mass closer to 0 and it is placed as a point mass at the point $c/\alpha$ for some number $c$. In this case, running Algorithm 2 without step 5 will result is implementing an auction with reserve price that is very close to $c/\alpha$. The probability though that the true agent with distribution $\mathcal{D}^*$ will buy this item goes to zero with a rate $\exp(-c/\alpha)$ as $c \to \infty$. Hence, the total revenue will be at most $(c/\alpha) \cdot \exp(-c/\alpha)$ and therefore we can make the total revenue to go to zero as we increase $c \to \infty$. Observe that this counterexample works even though we assumed that the initial distribution $\mathcal{D}^*$ is MHR.

We next provide the analysis of the performance of Algorithm 2 for MHR and regular distributions. The proof of the following result can be found in Appendix D.

**Theorem 4.3** (Finite samples, Regular distribution). *Let $\mathbf{D}^* = \mathcal{D}_1^* \times \cdots \times \mathcal{D}_n^*$ be a product distribution where every $\mathcal{D}_i^*$ is regular. Let also $\tilde{\mathbf{D}} = \tilde{\mathcal{D}}_1 \times \cdots \tilde{\mathcal{D}}_n$ be any product distribution such that for all $i \in [n]$ it holds that $d_k(\mathcal{D}_i^*, \tilde{\mathcal{D}}_i) \leqslant \alpha_i$. If $\tilde{M}$ is the mechanism that Algorithm 2 outputs with input $m$ samples from $\tilde{\mathbf{D}}$ and assume that $m = \tilde{\Omega}\left(\max_{i\in[n]}\left\{\log(\frac{1}{\delta})/\alpha_i^2\right\}\right)$ then it holds that*

$$\Pr\left(\mathrm{Rev}(\tilde{M}, \mathbf{D}^*) \geqslant \left(1 - O\left(\sqrt{\sum_{i=1}^n \alpha_i}\right)\right) \cdot \mathrm{OPT}(\mathbf{D}^*)\right) \geqslant 1 - \delta.$$

*Additionally, in the single-bidder case with $n = 1$ and $\alpha = \alpha_1$ the sample requirement becomes $m = \tilde{\Omega}\left(\log(\frac{1}{\delta})/\alpha^{3/2}\right)$.*

The corresponding theorem for MHR distributions is the following, whose proof can be found in Appendix D.

**Theorem 4.4** (Finite samples, MHR distribution). *Let $\mathbf{D}^* = \mathcal{D}_1^* \times \cdots \times \mathcal{D}_n^*$ be a product distribution where every $\mathcal{D}_i^*$ is MHR. Let also $\tilde{\mathbf{D}} = \tilde{\mathcal{D}}_1 \times \cdots \tilde{\mathcal{D}}_n$ be any product distribution such that for all $i \in [n]$ it holds that $d_k(\mathcal{D}_i^*, \tilde{\mathcal{D}}_i) \leqslant \alpha_i$. If $\tilde{M}$ is the mechanism that Algorithm 2 outputs with input $m$ samples from $\tilde{\mathbf{D}}$ and assume that $m = \tilde{\Omega}\left(\max_{i\in[n]}\left\{\log\left(\frac{1}{\delta}\right)/\alpha_i^2\right\}\right)$ then it holds that*

$$\Pr\left(\mathrm{Rev}(\tilde{M}, \mathbf{D}^*) \geqslant \left(1 - \tilde{O}\left(\sum_{i=1}^n \alpha_i\right)\right) \cdot \mathrm{OPT}(\mathbf{D}^*)\right) \geqslant 1 - \delta.$$

We make a few remarks about the sample complexity upper bounds in the sequel.

First, in both Theorem 4.3 and Theorem 4.4, the sample complexity upper bounds depend in a simple way on the sum of all the fractions of corruptions for each bidder; i.e., $\sum_{i=1}^n \alpha_i$, indicating the important effect of the *total* amount of corruption. Second, for regular distributions, in Theorem 4.3 we obtain a tight sample complexity bound for the single-bidder case, with $m = \tilde{\Omega}\left(\log(\frac{1}{\delta})/\alpha^{3/2}\right)$. For multi-bidder settings, our upper bound contains a small gap, with $m = \tilde{\Omega}\left(\max_{i\in[n]}\left\{\log(\frac{1}{\delta})/\alpha_i^2\right\}\right)$. Whether such a gap can be matched is an interesting open question for future work. Lastly, comparing Theorem 4.3 and Theorem 4.4, it appears that for the multi-bidder settings the sample complexity bounds are of the same order, but we emphasize the key difference that for regular distributions this sample size is needed to provide a much *weaker* guarantee on the revenue objective, which is a $\left(1 - O\left(\sqrt{\sum_{i=1}^n \alpha_i}\right)\right)$ fraction of the optimal revenue, while the guarantee for MHR distributions is a $(1 - O(\sum_{i=1}^n \alpha_i))$ fraction of the optimal revenue.

We next provide an information-theoretic lower bound that establishes the tightness of our upper bounds for the single-bidder single-item case with regular and MHR distributions.

**Theorem 4.5** (Sample complexity lower bounds). *Let $M$ be any DSIC and IR mechanism for a single-item single-buyer setting that takes as input $m$ samples from a distribution $\tilde{\mathcal{D}}$. If*

$$\mathrm{Rev}(M, \mathcal{D}^*) \geqslant (1 - O(\sqrt{\alpha})) \cdot \mathrm{OPT}(\mathcal{D}^*),$$

*for all distributions $\mathcal{D}^*$ such that $d_k(\mathcal{D}^*, \tilde{\mathcal{D}}) \leqslant \alpha$, where $\mathcal{D}^*$ is regular, then $m \geqslant \tilde{\Omega}\left(\log(\frac{2}{\delta})/\alpha^{3/2}\right)$. Additionally, if*

$$\mathrm{Rev}(M, \mathcal{D}^*) \geqslant (1 - O(\alpha)) \cdot \mathrm{OPT}(\mathcal{D}^*),$$

*for all distributions $\mathcal{D}^*$ such that $d_k(\mathcal{D}^*, \tilde{\mathcal{D}}) \leqslant \alpha$, where $\mathcal{D}^*$ is MHR, we have $m \geqslant \tilde{\Omega}\left(\log(\frac{2}{\delta})/\alpha^{3/2}\right)$.*

Theorem 4.5 provides a general sample complexity lower bound on learning a near-optimal auction with at least a $(1 - O(\sqrt{n \cdot \alpha}))$ fraction of the optimal revenue under the true valuation distribution. In comparison to our upper bounds (see Theorem 4.3 and Theorem 4.4), there is a small gap and we leave the nature of this gap as an open question for future work.

## 5 Conclusions

We have studied the learning of revenue-optimal auctions for multiple bidders, in a setting in which the samples can be corrupted adversarially. We first consider the information-theoretic limit in a

population model, assuming exact knowledge of the adversarially perturbed valuation distribution. We develop a theoretical algorithm which obtains a tight upper bound on the revenue for the MHR and regular distributions, obtaining the information-theoretic limit of the robustness guarantee. We then relax the population model and derive sample complexity bounds for learning optimal auctions from samples. We propose a practical algorithm which takes the corrupted samples as input, and provide the sample complexity upper bounds for the MHR distribution case and the single-bidder regular distribution case. We also provide accompanying sample complexity lower bounds, and demonstrate a small gap relative to the corresponding upper bounds.

## Acknowledgments and funding transparency statement.

We wish to acknowledge support from the Vannevar Bush Faculty Fellowship program under grant number N00014-21-1-2941. WG also acknowledges support from a Google PhD fellowship. MZ was supported by NSF DMS-2023505 (FODSI).

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
