# A   Proofs of Technical Lemmas

**Lemma 3.4.** A distribution with CDF $F$ is MHR if and only if $h_M(x; F)$ is a convex function of $x$. Similarly, $F$ is regular if and only if $h_r(x; F)$ is a convex function of $x$. Moreover, for two MHR (resp. regular) distributions $F_1$ and $F_2$, such that $F_1 \succeq F_2$, then we have that $h_M(x; F_1) \leqslant h_M(x; F_2)$ (resp. $h_r(x; F_1) \leqslant h_r(x; F_2)$) for all $x$.

*Proof.* We first show that given the CDF of any MHR distribution $F(x) : \mathbb{R}_+ \to [0, 1]$, $h_M(x) \overset{\text{def}}{=} -\log(1 - F(x))$ is a convex, non-decreasing function with $h(0) = 0$. (Without loss of generality, we consider $x \in [0, \infty]$, i.e. $\arg\min_x h(x) = 0$.) We first present the analysis for the case when the distribution is continuous and smooth, and then generalize the same statement to discrete distributions.

MHR continuous distributions:

Denote the corresponding PDF of $F(x)$ as $f(x)$, and $g(x) \overset{\text{def}}{=} \frac{f(v)}{1-F(v)}$. By definition, $F(0) = 0$ implies $h_M(0) = 0$. Then, given that $F(x)$ is MHR, we have that $g(x)$ is monotone non-decreasing. By construction,

$$(h_M(x))'' = \left( \frac{f(v)}{1 - F(v)} \right)' = g'(x) \geqslant 0.$$

Therefore, $h_M(x)$ is convex. Moreover, since $F(x)$ is a CDF thus non-decreasing, $h_M(x) = -\log(1 - F(x))$ is also non-decreasing. We show that given any $h_M(x) : \mathbb{R}_+ \to \mathbb{R}_+$, such that $h_M(x)$ is convex, non-decreasing, $h_M(0) = 0$, and $\max_x h_M(x) = \infty$. Then, $F(x) \overset{\text{def}}{=} 1 - \exp(-h_M(x))$ is CDF of an MHR distribution.

By construction, $h_M(0) = 0$ implies $F(0) = 0$, and $\max_x h_M(x)$ implies $\max_x F(x) = 1$. Also given that $h_M(x)$ is convex, $g'(x) = \left( \frac{f(v)}{1-F(v)} \right)' = (h_M(x))'' \geqslant 0$, which by definition implies $F(x)$ is MHR.

MHR discrete distributions:

The lemma statement generalizes to the case when the valuation is discrete. We assume that the valuation can take a discrete set of values $\{x_i\}, i = 1, \cdots, n$. Without loss of generality, we will restrict these values to the set $\mathbb{N}_0$ with probability mass function $P(x = i) = p_i; i = 0 \cdots n$. We define the *discrete* hazard rate as:

$$g(x_i) = \frac{P(x = i)}{P(x \geqslant i)}.$$

Then, the valuation distribution is MHR iff the discrete hazard rate is non-decreasing:

$$g(x_{i+1}) \geqslant g(x_i), \tag{2}$$

for all $i = 0 \cdots n$.

In this case, our link function will also be discrete. Further, denote $s_i \overset{\text{def}}{=} P(x \geqslant i)$, then

$$h(x_i) = -\log(P(x \geqslant x_i)) = -\log(s_i).$$

Then $h(x)$ is convex if and only if for any $i \geqslant 0$,

$$h(x_{i+2}) - h(x_{i+1}) \geqslant h(x_{i+1} - h(x_i). \tag{3}$$

We show that Eq (2) and Eq (3) are equivalent. Notice that

$$h(x_{i+2}) - h(x_{i+1}) \geqslant h(x_{i+1}) - h(x_i)$$
$$\iff \frac{s_{i+1}}{s_{i+1} - p_{i+1}} \geqslant \frac{s_i}{s_i - p_i}$$
$$\iff p_{i+1}s_i \geqslant p_i s_{i+1}$$
$$\iff \frac{p_{i+1}}{s_{i+1}} \geqslant \frac{p_i}{s_i}$$
$$\iff g(x_{i+1}) \geqslant g(x_i),$$

which completes the proof.

Regular continuous distributions:

We further prove a similar statement for regular continuous distributions. First, given a CDF of a regular distribution $F(x)$,

$$\left(\frac{1}{1-F(x)}\right)'' = \frac{(1-F(x))f(x)' + 2f(x)^2}{(1-F(x))^3}.$$

By definition, the virtual value function is $\phi(x) \overset{\text{def}}{=} v - \frac{1-F(x)}{f(x)}$, and

$$\phi'(x) = \frac{(1-F(x))f(x)' + 2f(x)^2}{f(x)^2}.$$

Therefore, $\left(\frac{1}{1-F(x)}\right)''$ and $\phi'(x)$ share the same sign. Moreover, the distribution with CDF as $F(x)$ is regular if and only if the virtual value $\phi(x)$ is monotonically non-decreasing, which is $\phi'(x) \geqslant 0$. Hence the regularity of $F(x)$ implies that $h_r(x) \overset{\text{def}}{=} \frac{1}{1-F(x)}$ is convex. Since $F(x)$ is a CDF thus non-decreasing, $h_r(x) = \frac{1}{1-F(x)}$ is also non-decreasing.

Regular discrete distributions:

Similar to the MHR distributions, the lemma statement generalizes to the case when the valuation is discrete for regular distributions. Assume that the valuation can take a discrete set of values $\{x_i\}, i = 1, \cdots, n$. Without loss of generality, we will restrict these values to the set $\mathbb{N}_0$ with probability mass function $P(x = i) = p_i; i = 0 \cdots n$. Further, consistent with the proof for MHR distributions, we denote $s_i \overset{\text{def}}{=} P(x \geqslant i)$.

The *discrete* virtual value function is defined as:

$$\phi(x_i) = x_i - \frac{s_i}{p_i},$$

and the valuation distribution is regular iff $\phi(x)$ is non-decreasing:

$$\phi(x_{i+1}) \geqslant \phi(x_i), \tag{4}$$

for all $i = 0 \cdots n$.

In this case, our link function will again be discrete:

$$h(x_i) = \frac{1}{P(x \geqslant x_i)} = \frac{1}{s_i}.$$

and $h(x)$ is convex if and only if for any $i \geqslant 0$,

$$h(x_{i+2}) - h(x_{i+1}) \geqslant h(x_{i+1}) - h(x_i). \tag{5}$$

We show that Eq (4) and Eq (5) are equivalent.

$$h(x_{i+2}) - h(x_{i+1}) \geqslant h(x_{i+1}) - h(x_i)$$
$$\iff \frac{1}{s_{i+2}} + \frac{1}{s_i} \geqslant \frac{2}{s_{i+1}}$$
$$\iff \frac{1}{s_{i+1} - p_{i+1}} + \frac{1}{s_i} \geqslant \frac{2}{s_{i+1}} \tag{6}$$
$$\iff s_{i+1}^2 + p_i p_{i+1} \geqslant s_i s_{i+1} - s_i p_{i+1}.$$
$$\iff p_i p_{i+1} + p_{i+1} s_i + s_{i+1}(s_{i+1} - s_i) \geqslant 0$$
$$\iff p_i p_{i+1} + p_{i+1} s_i - s_{i+1} p_i \geqslant 0$$

Moreover, from the regularity condition Eq (4), we have

$$\phi(x_{i+1}) \geqslant \phi(x_i)$$
$$\iff i + 1 - \frac{s_{i+1}}{p_{i+1}} \geqslant i - \frac{s_i}{p_i}$$
$$\iff 1 - \frac{s_{i+1}}{p_{i+1}} + \frac{s_i}{p_i} \geqslant 0 \tag{7}$$
$$\iff p_i p_{i+1} + p_{i+1} s_i - s_{i+1} p_i \geqslant 0.$$

Combining (6) and (7) together completes the proof.

Stochastic dominance:

Lastly, we show that for two MHR (resp. regular) distributions $F_1$ and $F_2$, such that $F_1 \succeq F_2$, then we have that $h_M(x; F_1) \leqslant h_M(x; F_2)$ (resp. $h_r(x; F_1) \leqslant h_r(x; F_2)$) for all $x$. This follows directly from the monotonicity of the link functions and the definition of stochastic dominance (see Definition 3.2).

Recall that the link function $h_M(x; F)$ for MHR distributions is defined as $h_M(x; F) = -\ln(1 - F(x))$, and the link function $h_r(x; F)$ for regular distributions is defined as $h_r(x; F) = 1/(1 - F(x))$. Therefore, for two MHR (resp. regular) distributions $F_1$ and $F_2$, $F_1(x) < F_2(x)$ implies $h_M(x, F_1) < h_M(x, F_2)$ (resp. $h_r(x, F_1) < h_r(x, F_2)$), which completes the proof. ∎

**Lemma 4.2.** Let $f$ be a non-decreasing piecewise constant function with $k$ pieces, then $Conv(f)$ can be computed in time $\text{poly}(k)$ and is a piecewise linear function with $O(k)$ pieces.

*Proof.* Given that $f(x)$ is a non-decreasing piecewise constant function with $k$ pieces, we show that the following iterative procedure outputs its lower convex envelope $Conv(f)$, which can be computed in time $\text{poly}(k)$ and is a piecewise linear function with $O(k)$ pieces. Figure 3 provides an illustration of the construction according to this procedure.

---

**Procedure 1** Computing lower convex envelope for non-decreasing piecewise constant functions

1: **Input: a piecewise constant function** $f(x) : \mathbb{R} \to \mathbb{R}$ **with $k$ pieces.** Denote the left starting point of each piece and the end point as $x_0, \ldots, x_k$.
2: **Initialize:** $i \leftarrow 0, i' \leftarrow 0$.
3: **while** $i \leqslant k - 1$ **do**
4:     $\bar{x}_{i'} \leftarrow x_i, g(\bar{x}_{i'}) \leftarrow f(x_i)$.
5:     $i' \leftarrow i' + 1$.
6:     Compute $i \leftarrow \arg\min_{i < j \leqslant k} \frac{f(x_j) - f(x_i)}{x_j - x_i}$.
7: **end while**
8: $\bar{x}_{i'} \leftarrow x_i, g(\bar{x}_{i'}) \leftarrow f(x_i); k' \leftarrow i'$.
9: **Return: a piecewise linear function** $g(x) : \mathbb{R} \to \mathbb{R}$ **with $k' < k$ pieces.** The left starting points of each piece and the end points are $\bar{x}_0, \ldots, \bar{x}_{i'}$, with the corresponding function values as specified in the procedure.

---

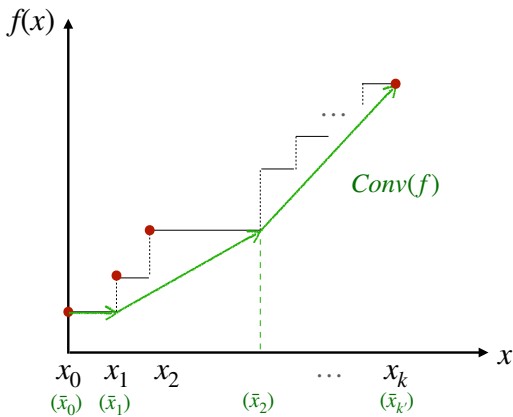

Figure 3: Lower convex envelope of a non-decreasing piecewise constant function $f(x)$.

First, the above procedure requires at most $k^2$ rounds. We show that its output, $g(x)$, is the lower convex envelope for $f(x)$. It is clear from construction that $g(x)$ is piecewise linear, with vertices at $\bar{x}_0, \ldots, \bar{x}_{k'}$. Moreover, $g(x) \leqslant f(x)$ for all $x$ by construction.

Next we show that $g(x)$ is convex. Consider at a round $t$ with $i = i_t, 1 < 1 < k$. Then, step (6) computes $i_{t+1} = \arg\min_{i_t < j \leqslant k} \frac{f(x_j) - f(x_{i_t})}{x_j - x_{i_t}}$. Further denote $\min_{i_t < j \leqslant k} \frac{f(x_j) - f(x_{i_t})}{x_j - x_{i_t}}$ as $s(i_t)$. We show that $s(i_{t+1}) \geqslant s_{i_t}$.

Suppose that $s(i_{t+1}) < s_{i_t}$. Then there exists $j^* > i_{t+1} > i_t$, such that

$$\frac{f(x_{j^*}) - f(x_{i_{t+1}})}{x_{j^*} - x_{i_{t+1}}} < \frac{f(x_{i_{t+1}}) - f(x_{i_t})}{x_{i_{t+1}} - x_{i_t}},$$

which further implies that

$$\frac{f(x_{j^*}) - f(x_{i_t})}{x_{j^*} - x_{i_t}} < \frac{f(x_{i_{t+1}}) - f(x_{i_t})}{x_{i_{t+1}} - x_{i_t}}.$$

Since $j^* > i_{t+1} > i_t$, this contradicts the fact that $i_{t+1} = \arg\min_{i_t < j \leqslant k} \frac{f(x_j) - f(x_{i_t})}{x_j - x_{i_t}}$. Therefore $s(i_{t+1}) \geqslant s_{i_t}$, which means that the slope of each piece for $g(x)$ is non-decreasing. Thus $g(x)$ is convex. Lastly, since $g(x)$ has all vertices with the same function values as $f(x)$, i.e. $g(x) = f(x)$ at all its vertices, and given that $g(x) \leqslant f(x)$ for all $x$, the values at these vertices are maximized and cannot be further improved. This completes the proof. ∎

We further provide two lemmas which present useful properties of the link functions in connection to the revenue.

**Lemma A.1.** *Given an MHR distribution with the CDF as $F(x) : \mathbb{R}_+ \to [0, 1]$. Define $h(x) \overset{\text{def}}{=} -\log(1 - F(x))$. Then, at any reserve price $x$, the expected revenue $R(x) = \exp(-h(x) + \log(x))$. Moreover, the optimal reserve price $P_F^*$ is the minimizer of $(h(x) - \log(x))$.*

*Proof.* First by construction, $h(x) - \log(x) = -\log(R(x))$. By definition, the optimal reserve price maximizes the revenue $R(x) = x(1 - F(x))$, thus

$$
\begin{aligned}
&\max \quad x(1 - F(x)) \\
\Longleftrightarrow \ &\min \quad -\log(x(1 - F(x))) \\
\Longleftrightarrow \ &\min \quad -\log(x) - \log(1 - F(x)) \\
\Longleftrightarrow \ &\min \quad h(x) - \log(x),
\end{aligned}
$$

which completes the proof. ∎

**Lemma A.2.** *Consider a valuation distribution $\mathcal{D}$ with CDF as $F(x)$. Denote the optimal reserve price as $P_F^*$ and the optimal expected revenue at $P_F^*$ as $\mathrm{OPT}_F$. Then $P_F^*$ should be $P_F^* \leqslant e$, assuming that $\mathrm{OPT}_F \leqslant 1$ and $F(x)$ is MHR.*

*Proof.* By Lemma A.1, $\mathrm{OPT}_F \leqslant 1$ implies that,

$$h(P_F^*) = \log(P_F^*) + b,$$

for some $b \geqslant 0$. Also by Lemma 3.4, $h$ is convex. Combined with the fact that $\mathrm{OPT}_F$ is the optimal reserve price and the concavity of $\log(x)$, $\mathrm{OPT}_F$ is the only point where $h(P_F^*) = \log(P_F^*) + b$ holds.

Now consider a linear function $y = ax, a > 0$, which is a tangent line of the function $\log(x) + b$. Denote the tangent point as $x^*$. Solving the equation that $a = (\log(x))' = \frac{1}{x}$, and $ax = \log(x) + b$ give that:

$$x^* = e^{1-b} \leqslant e.$$

Suppose that $P_F^* > x^*$. Consider the linear function $g(x) = \frac{h(P_F^*)}{P_F^*} x$. Since $x^*$ is the tangent point, there exists a point $\bar{x} < P_F^*$, such that $g(\bar{x}) = \log(\bar{x}) + b$. Further, since $h$ is convex, for any point $0 < x < P_F^*$, we have $h(x) < g(x)$. By the continuity of $\log(x)$ and $h(x)$, there exists $\bar{x}' < P_F^*$, such that $h(\bar{x}') = \log(\bar{x}) + b$. This implies that $\bar{x}'$ achieves a larger revenue than $P_F^*$, and contradicts the fact that $P_F^*$ is the optimal reserve price. Hence, $P_F^* < x^* \leqslant e$, which completes the proof. ∎

# B  Proof of Upper Bounds for the Population Model

We first prove the following technical lemma that connects the coordinate Kolmogorov distance with the difference in expectation of increasing functions.

**Definition B.1** (Increasing Functions and Sets)**.** Let $u : \mathbb{R}^n \to \mathbb{R}$, we say that $u$ is increasing if for every $\mathbf{v} = (v_1, \ldots, v_n)$, $\mathbf{v}' = (v'_1, \ldots, v'_n)$ such that $v'_i \geqslant v_i$, it holds that $u(\mathbf{v}') \geqslant u(\mathbf{v})$. We say that the subset $A \subseteq \mathbb{R}^n$ is increasing if and only if its characteristic function $\mathbf{1}_A(\mathbf{x})$ is an increasing function of $\mathbf{x}$.

**Lemma B.2.** *Let* $\mathbf{D} = \mathcal{D}_1 \times \cdots \times \mathcal{D}_n$, $\mathbf{D}' = \mathcal{D}'_1 \times \cdots \times \mathcal{D}'_n$ *be product $n$-dimensional distributions with* $d_k(\mathcal{D}_i, \mathcal{D}'_i) \leqslant \alpha_i$. *Then for every increasing function* $u : \mathbb{R}^n \to [0, \bar{u}]$ *it holds that*

$$\left| \mathop{\mathbb{E}}_{\mathbf{v} \sim \mathbf{D}}[u(\mathbf{v})] - \mathop{\mathbb{E}}_{\mathbf{v}' \sim \mathbf{D}'}[u(\mathbf{v}')] \right| \leqslant \bar{u} \cdot \left( \sum_{i=1}^{n} \alpha_i \right).$$

*Proof.* Our first step is to prove that the lemma holds for any function $u$ that is a characteristic function of an increasing set $A$ and then we extend to all increasing functions.

Let $u = \mathbf{1}_A$ we have that $\mathbb{E}_{\mathbf{v} \sim \mathbf{D}}[u(\mathbf{v})] = \Pr_{\mathbf{v} \sim \mathbf{D}}(\mathbf{v} \in A)$. We define the sequence of distributions $\mathbf{D}_j = \mathcal{D}'_1 \times \cdots \times \mathcal{D}'_j \times \mathcal{D}_{j+1} \times \cdots \times \mathcal{D}_n$ for $j = 0, \ldots, n$, where obviously $\mathbf{D}_0 = \mathbf{D}$ and $\mathbf{D}_n = \mathbf{D}'$. Now via triangle inequality we have that

$$\left| \mathop{\Pr}_{\mathbf{v} \sim \mathbf{D}} (\mathbf{v} \in A) - \mathop{\Pr}_{\mathbf{v} \sim \mathbf{D}'} (\mathbf{v} \in A) \right| \leqslant \sum_{j=1}^{n} \left| \mathop{\Pr}_{\mathbf{v} \sim \mathbf{D}_j} (\mathbf{v} \in A) - \mathop{\Pr}_{\mathbf{v} \sim \mathbf{D}_{j-1}} (\mathbf{v} \in A) \right|. \tag{8}$$

Let $b_j(\mathbf{v}_{-j})$ be the threshold of the step function $\mathbf{1}_A(v_j, \mathbf{v}_{-j})$ when we fix $\mathbf{v}_{-j}$ and we view it as a function of $v_j$. Now we have that

$$\mathop{\Pr}_{\mathbf{v} \sim \mathbf{D}_j} (\mathbf{v} \in A) = \int_{\mathbb{R}^n} \mathbf{1}_A(x_j, \mathbf{x}_{-j}) \ d\mathcal{D}'_1(x_1) \cdots d\mathcal{D}'_j(x_j) \cdot d\mathcal{D}_{j+1}(x_{j+1}) \cdots d\mathcal{D}_n(x_n)$$

$$= \int_{\mathbb{R}^{n-1}} (1 - \mathcal{D}'_j(b_j(\mathbf{x}_{-j}))) \ d\mathcal{D}'_1(x_1) \cdots d\mathcal{D}'_{j-1}(x_{j-1}) \cdot d\mathcal{D}_{j+1}(x_{j+1}) \cdots d\mathcal{D}_n(x_n)$$

similarly we have

$$\mathop{\Pr}_{\mathbf{v} \sim \mathbf{D}_{j-1}} (\mathbf{v} \in A) = \int_{\mathbb{R}^{n-1}} (1 - \mathcal{D}_j(b_j(\mathbf{x}_{-j}))) \ d\mathcal{D}'_1(x_1) \cdots d\mathcal{D}'_{j-1}(x_{j-1}) \cdot d\mathcal{D}_{j+1}(x_{j+1}) \cdots d\mathcal{D}_n(x_n).$$

Combining these we get that

$$\left| \mathop{\Pr}_{\mathbf{v} \sim \mathbf{D}_j} (\mathbf{v} \in A) - \mathop{\Pr}_{\mathbf{v} \sim \mathbf{D}_{j-1}} (\mathbf{v} \in A) \right| \leqslant$$

$$\leqslant \int_{\mathbb{R}^{n-1}} \left| \mathcal{D}'_j(b_j(\mathbf{x}_{-j})) - \mathcal{D}_j(b_j(\mathbf{x}_{-j})) \right| \ d\mathcal{D}'_1(x_1) \cdots d\mathcal{D}'_{j-1}(x_{j-1}) \cdot d\mathcal{D}_{j+1}(x_{j+1}) \cdots d\mathcal{D}_n(x_n).$$

from the latter we can use the fact that $d_k(\mathcal{D}_j, \mathcal{D}'_j) \leqslant \alpha_j$ and we get that

$$\left| \mathop{\Pr}_{\mathbf{v} \sim \mathbf{D}_j} (\mathbf{v} \in A) - \mathop{\Pr}_{\mathbf{v} \sim \mathbf{D}_{j-1}} (\mathbf{v} \in A) \right| \leqslant \alpha_j.$$

Applying the above to (8) we get that

$$\left| \mathop{\Pr}_{\mathbf{v} \sim \mathbf{D}} (\mathbf{v} \in A) - \mathop{\Pr}_{\mathbf{v} \sim \mathbf{D}'} (\mathbf{v} \in A) \right| \leqslant \sum_{j=1}^{n} \alpha_j. \tag{9}$$

The last steps is to extend the above to arbitrary increasing functions. We are going to approximate the increasing function $u$ via a sequence of functions $u_k$ which uniformly converges to $u$. Then we will show the statement of the lemma for every function $u_k$ which by uniform convergence implies the lemma for $u$ as well. We set $A_{i,k} \triangleq \{ \mathbf{x} \in \mathbb{R}^n \mid u(\mathbf{x}) \geqslant \frac{i}{k} \bar{u} \}$ and we define

$$u_k(\mathbf{x}) = \frac{\bar{u}}{k} \sum_{i=1}^{k} \mathbf{1}_{A_{i,k}}(\mathbf{x}).$$

Observe from the above definition that $u_k \to u$ uniformly and since $u$ is increasing we also have that all the sets $A_i$ are increasing. Also observe that

$$\mathop{\mathbb{E}}_{\mathbf{v} \sim \mathbf{D}}[u_k(\mathbf{v})] = \frac{\bar{u}}{k} \sum_{i=1}^{k} \mathop{\Pr}_{\mathbf{v} \sim \mathbf{D}}(\mathbf{v} \in A_{i,k})$$

therefore we get that

$$\left| \mathop{\mathbb{E}}_{\mathbf{v} \sim \mathbf{D}}[u_k(\mathbf{v})] - \mathop{\mathbb{E}}_{\mathbf{v} \sim \mathbf{D}'}[u_k(\mathbf{v})] \right| \leqslant \frac{\bar{u}}{k} \sum_{i=1}^{k} \left| \mathop{\Pr}_{\mathbf{v} \sim \mathbf{D}}(\mathbf{v} \in A_{i,k}) - \mathop{\Pr}_{\mathbf{v} \sim \mathbf{D}'}(\mathbf{v} \in A_{i,k}) \right|.$$

Now we can apply (9) and we get

$$\left| \mathop{\mathbb{E}}_{\mathbf{v} \sim \mathbf{D}}[u_k(\mathbf{v})] - \mathop{\mathbb{E}}_{\mathbf{v} \sim \mathbf{D}'}[u_k(\mathbf{v})] \right| \leqslant \bar{u} \cdot \left( \sum_{j=1}^{n} \alpha_j \right).$$

Finally, since this is true for every $u_k$ and $u$ converges uniformly to $u$ the above should be true for $u$ as well and hence the lemma follows. ∎

We are going to use Lemma B.2 both for the regular distributions case and for the MHR distributions case.

## B.1 Monotone Hazard Rate Distributions—Proof of Theorem 3.6

In this section we show the part of the Theorem 3.6 related to $n > 1$. For the stronger result for the case $n = 1$ we refer to Section B.3.

Let $\tilde{\mathbf{D}}$ be the corrupted product distribution that we observe, $\widehat{\mathbf{D}}$ be the output distribution of Algorithm 1, $\mathbf{D}^*$ be the original distribution that we are interested in. We know from the description of Algorithm 1 for $\widehat{\mathbf{D}} = \widehat{\mathcal{D}}_1 \times \cdots \times \widehat{\mathcal{D}}_n$ that $\widehat{\mathcal{D}}_i$ is MHR, that $d_k(\widehat{\mathcal{D}}_i, \mathcal{D}_i^*) \leqslant \alpha_i$ and that $\widehat{\mathcal{D}}_i \preceq \mathcal{D}_i^*$. We also know that $\mathcal{D}_i^*$ is MHR. Finally, we know that the output $M$ of Algorithm 1 is the Myerson optimal mechanism for the distribution $\widehat{\mathbf{D}}$ and hence $\mathrm{Rev}(M, \widehat{\mathbf{D}}) = \mathrm{OPT}(\widehat{\mathbf{D}})$. So applying the strong revenue monotonicity lemma 3.3 we have that

$$\mathrm{OPT}(\widehat{\mathbf{D}}) = \mathrm{Rev}(M, \widehat{\mathbf{D}}) \leqslant \mathrm{Rev}(M, \mathbf{D}^*). \tag{10}$$

Therefore to show Theorem 3.6, it suffices to show that

$$\mathrm{OPT}(\widehat{\mathbf{D}}) \geqslant \left( 1 - \tilde{O}\left( \sum_{i=1}^{n} \alpha_i \right) \right) \cdot \mathrm{OPT}(\mathbf{D}^*). \tag{11}$$

We are going to use the following result from Cai and Daskalakis [2011] but with the formulation obtained in Lemma 17 of Guo et al. [2019], combined with the weak revenue monotonicity (Lemma 3 of Guo et al. [2019]).

**Theorem B.3** (Cai and Daskalakis [2011]). *For any product MHR distribution $\mathbf{D}$, and any $\frac{1}{4} \geqslant \varepsilon \geqslant 0$ and $u \geqslant c \cdot \log\left(\frac{1}{\varepsilon}\right) \mathrm{OPT}(\mathbf{D})$. Let $t_u(\mathcal{D}_1)$, ..., $t_u(\mathcal{D}_n)$ be the distributions obtained by truncating $\mathcal{D}_1, \ldots, \mathcal{D}_n$ at the value $\bar{u}$ and let $t_u(\mathbf{D})$ be their product distribution, where $c$ is an absolute constant. Then, we have that*

$$\mathrm{OPT}(\mathbf{D}) \geqslant \mathrm{OPT}(t_u(\mathbf{D})) \geqslant (1 - \varepsilon) \cdot \mathrm{OPT}(\mathbf{D}).$$

Now let $\bar{u} = c \cdot \log\left(\frac{1}{\varepsilon}\right) \mathrm{OPT}(\mathbf{D}^*)$, then we also have that $\bar{u} \geqslant c \cdot \log\left(\frac{1}{\varepsilon}\right) \mathrm{OPT}(\widehat{\mathbf{D}})$ due to weak revenue monotonicity (Lemma 3 of Guo et al. [2019]). Hence, applying Theorem B.3 we have that

$$\mathrm{OPT}(\widehat{\mathbf{D}}) \geqslant \mathrm{OPT}(t_{\bar{u}}(\widehat{\mathbf{D}})) \qquad \text{and} \qquad \mathrm{OPT}(t_{\bar{u}}(\mathbf{D}^*)) \geqslant (1 - \varepsilon) \cdot \mathrm{OPT}(\mathbf{D}^*). \tag{12}$$

Since we know that $d_k(\widehat{\mathcal{D}}_i, \mathcal{D}_i^*) \leqslant \alpha_i$ we also have that $d_k(t_{\bar{u}}(\widehat{\mathcal{D}}_i), t_{\bar{u}}(\mathcal{D}_i^*)) \leqslant \alpha_i$. Let now $M_{\bar{u}}^*$ be the optimal mechanism for the distribution $t_{\bar{u}}(\mathbf{D}^*)$. It is easy to see that the ex-post revenue obtained

from the mechanism $M_{\bar{u}}^*$ is an increasing function of the observed bids. Hence, we can apply Lemma B.2 to the $[0, \bar{u}]$ bounded distributions $t_{\bar{u}}(\widehat{\mathbf{D}})$ and $t_{\bar{u}}(\mathbf{D}^*)$ and we get that

$$\text{OPT}(t_{\bar{u}}(\widehat{\mathbf{D}})) \geqslant \text{Rev}(M_{\bar{u}}^*, t_{\bar{u}}(\widehat{\mathbf{D}})) \geqslant \text{Rev}(M_{\bar{u}}^*, t_{\bar{u}}(\mathbf{D}^*)) - \bar{u} \cdot \left( \sum_{i=1}^n \alpha_i \right)$$

$$= \text{OPT}(t_{\bar{u}}(\mathbf{D}^*)) - \bar{u} \cdot \left( \sum_{i=1}^n \alpha_i \right). \tag{13}$$

If we combine (12) and (13) then we have that

$$\text{OPT}(\widehat{\mathbf{D}}) \geqslant (1 - \varepsilon) \cdot \text{OPT}(\mathbf{D}^*) - \bar{u} \cdot \left( \sum_{i=1}^n \alpha_i \right). \tag{14}$$

Now we can substitute the value of $\bar{u}$ to the above inequality and we get that

$$\text{OPT}(\tilde{\mathbf{D}}) \geqslant \left( 1 - c \cdot \log\left(\frac{1}{\varepsilon}\right) \cdot \left( \sum_{i=1}^n \alpha_i \right) - \varepsilon \right) \cdot \text{OPT}(\mathbf{D}).$$

Finally, setting $\varepsilon = \sum_{i=1}^n \alpha_i$ we get

$$\text{OPT}(\tilde{\mathbf{D}}) \geqslant \left( 1 - (c+1) \cdot \left( \sum_{i=1}^n \alpha_i \right) \cdot \log\left( \frac{1}{\sum_{i=1}^n \alpha_i} \right) \right) \cdot \text{OPT}(\mathbf{D}).$$

Hence, (11) follows and as we explained this proves Theorem 3.6.

## B.2   Regular Distributions—Proof of Theorem 3.8

Let $\tilde{\mathbf{D}}$ be the corrupted product distribution that we observe, $\widehat{\mathbf{D}}$ be the output distribution of Algorithm 1, $\mathbf{D}^*$ be the original distribution that we are interested in. We know from the description of Algorithm 1 for $\widehat{\mathbf{D}} = \widehat{\mathcal{D}}_1 \times \cdots \times \widehat{\mathcal{D}}_n$ that $\widehat{\mathcal{D}}_i$ is a regular distribution, that $d_k(\widehat{\mathcal{D}}_i, \mathcal{D}_i^*) \leqslant \alpha_i$ and that $\widehat{\mathcal{D}}_i \preceq \mathcal{D}_i^*$. We also know that $\mathcal{D}_i^*$ is regular. Finally, we know that the output $M$ of Algorithm 1 is the Myerson optimal mechanism for the distribution $\widehat{\mathbf{D}}$ and hence $\text{Rev}(M, \widehat{\mathbf{D}}) = \text{OPT}(\widehat{\mathbf{D}})$. So applying the strong revenue monotonicity lemma 3.3 we have that

$$\text{OPT}(\widehat{\mathbf{D}}) = \text{Rev}(M, \widehat{\mathbf{D}}) \leqslant \text{Rev}(M, \mathbf{D}^*). \tag{15}$$

Therefore to show Theorem 3.8, it suffices to show that

$$\text{OPT}(\widehat{\mathbf{D}}) \geqslant \left( 1 - \tilde{O}\left( \sum_{i=1}^n \alpha_i \right) \right) \cdot \text{OPT}(\mathbf{D}^*). \tag{16}$$

We are going to use the following theorem from Devanur et al. [2016], combined with the weak revenue monotonicity (Lemma 3 of Guo et al. [2019]).

**Theorem B.4** (Lemma 2 of Devanur et al. [2016]). *Let $\mathbf{D}$ be a product of $n$ regular distributions and $\text{OPT}(\mathbf{D})$ be the optimal revenue of $\mathbf{D}$. Suppose $\frac{1}{4} \geqslant \varepsilon \geqslant 0$ and $u \geqslant \frac{1}{\varepsilon}\text{OPT}(\mathbf{D})$. Let $t_u(\mathcal{D}_1)$, ..., $t_u(\mathcal{D}_n)$ be the distributions obtained by truncating $\mathcal{D}_1$, ..., $\mathcal{D}_n$ at the value $u$ and let $t_u(\mathbf{D})$ be their product distribution. Then, we have that*

$$\text{OPT}(\mathbf{D}) \geqslant \text{OPT}(t_u(\mathbf{D})) \geqslant (1 - 4\varepsilon) \cdot \text{OPT}(\mathbf{D}).$$

Now let $\bar{u} = \frac{1}{\varepsilon}\text{OPT}(\mathbf{D}^*)$, then we also have that $\bar{u} \geqslant \frac{1}{\varepsilon}\text{OPT}(\widehat{\mathbf{D}})$ due to weak revenue monotonicity (Lemma 3 of Guo et al. [2019]). Hence, applying Theorem B.4 we have that

$$\text{OPT}(\widehat{\mathbf{D}}) \geqslant \text{OPT}(t_{\bar{u}}(\widehat{\mathbf{D}})) \qquad \text{and} \qquad \text{OPT}(t_{\bar{u}}(\mathbf{D}^*)) \geqslant (1 - \varepsilon) \cdot \text{OPT}(\mathbf{D}^*). \tag{17}$$

Since we know that $d_k(\widehat{\mathcal{D}}_i, \mathcal{D}_i^*) \leqslant \alpha_i$ we also have that $d_k(t_{\bar{u}}(\widehat{\mathcal{D}}_i), t_{\bar{u}}(\mathcal{D}_i^*)) \leqslant \alpha_i$. Let now $M_{\bar{u}}^*$ be the optimal mechanism for the distribution $t_{\bar{u}}(\mathbf{D}^*)$. It is easy to see that the ex-post revenue obtained

from the mechanism $M_{\bar{u}}^*$ is an increasing function of the observed bids. Hence, we can apply Lemma B.2 to the $[0, \bar{u}]$ bounded distributions $t_{\bar{u}}(\widehat{\mathbf{D}})$ and $t_{\bar{u}}(\mathbf{D}^*)$ and we get that

$$\mathrm{OPT}(t_{\bar{u}}(\widehat{\mathbf{D}})) \geqslant \mathrm{Rev}(M_{\bar{u}}^*, t_{\bar{u}}(\widehat{\mathbf{D}})) \geqslant \mathrm{Rev}(M_{\bar{u}}^*, t_{\bar{u}}(\mathbf{D}^*)) - \bar{u} \cdot \left(\sum_{i=1}^{n} \alpha_i\right)$$

$$= \mathrm{OPT}(t_{\bar{u}}(\mathbf{D}^*)) - \bar{u} \cdot \left(\sum_{i=1}^{n} \alpha_i\right). \qquad (18)$$

If we combine (17) and (18) then we have that

$$\mathrm{OPT}(\widehat{\mathbf{D}}) \geqslant (1 - \varepsilon) \cdot \mathrm{OPT}(\mathbf{D}^*) - \bar{u} \cdot \left(\sum_{i=1}^{n} \alpha_i\right). \qquad (19)$$

Now we can substitute the value of $\bar{u}$ to the above inequality and we get that

$$\mathrm{OPT}(\tilde{\mathbf{D}}) \geqslant \left(1 - \frac{1}{\varepsilon} \cdot \left(\sum_{i=1}^{n} \alpha_i\right) - 4\varepsilon\right) \cdot \mathrm{OPT}(\mathbf{D}).$$

Finally, setting $\varepsilon = \sqrt{\sum_{i=1}^{n} \alpha_i}$ we get

$$\mathrm{OPT}(\tilde{\mathbf{D}}) \geqslant \left(1 - 5 \cdot \sqrt{\sum_{i=1}^{n} \alpha_i}\right) \cdot \mathrm{OPT}(\mathbf{D}).$$

Hence, (16) follows and as we explained this proves Theorem 3.8.

### B.3 MHR Distributions – Proof of Theorem 3.6, $n = 1$ Case

In this subsection we show the part of the Theorem 3.6 related to $n = 1$, for which we obtain a stronger result compared to the case $n > 1$. We first show a useful proposition:

**Proposition B.5.** *Consider two MHR distributions $\mathcal{D}_1$, $\mathcal{D}_2$ with CDFs as $F_1$ and $F_2$, such that $d_k(\mathcal{D}_1, \mathcal{D}_1) \leqslant \alpha$, and $F_1(x) \geqslant F_2(x)$ for all $x \in \mathbb{R}_+$. Denote the optimal expected revenue under $\mathcal{D}_1$ and $\mathcal{D}_2$ as $\mathrm{OPT}_{F_1}$ and $\mathrm{OPT}_{F_2}$, and the corresponding optimal reserve prices as $P_{F_1}^*$ and $P_{F_2}^*$. Then,*

$$(1 + \alpha e)^{-1} \leqslant \frac{\mathrm{OPT}_{F_1}}{\mathrm{OPT}_{F_2}} \leqslant 1 + \alpha e.$$

*Proof.* Consider two MHR distributions $\mathcal{D}_1$, $\mathcal{D}_2$ with CDFs as $F_1$ and $F_2$, such that $d_k(\mathcal{D}_1, \mathcal{D}_1) \leqslant \alpha$, and $F_1(x) \geqslant F_2(x)$ for all $x \in \mathbb{R}_+$. Denote the optimal expected revenue under $\mathcal{D}_1$ and $\mathcal{D}_2$ as $\mathrm{OPT}_{F_1}$ and $\mathrm{OPT}_{F_2}$, and the corresponding optimal reserve prices as $P_{F_1}^*$ and $P_{F_2}^*$. Without loss of generality, we consider $\mathrm{OPT}_{F_1} \geqslant \mathrm{OPT}_{F_2}$. Further, since the ratio of the revenues, e.g. $\frac{\mathrm{OPT}_{F_1}}{\mathrm{OPT}_{F_2}}$ is scale invariant, we assume without loss of generality that $\mathrm{OPT}_{F_1} = 1$.

By Lemma A.2, we have $P_{F_1}^* \leqslant e$. By Lemma A.1, $\mathrm{OPT}_{F_1} = 1$ implies that $h_1(P_{F_1}^*) = \log(P_{F_1}^*)$. Since $P_{F_1}^* \leqslant e$, we have

$$h_1(P_{F_1}^*) \leqslant 1$$
$$\iff -\log(1 - F_1(P_{F_1}^*)) \leqslant 1$$
$$\iff F_1(P_{F_1}^*) \leqslant 1 - \frac{1}{e}$$
$$\iff 1 - F_1(P_{F_1}^*) \geqslant \frac{1}{e}.$$

Therefore, since $F_1$ is non-decreasing, for any $x < P^*_{F_1}$, $1 - F_1(x) \geqslant \frac{1}{e}$. So for any $x < P^*_{F_1}$, we have

$$
\begin{aligned}
|h_1(x) - h_2(x)| &= \left| \log \left( \frac{1 - F_2(x)}{1 - F_1(x)} \right) \right| \\
&= \left| \log \left( 1 + \frac{F_1(x) - F_2(x)}{1 - F_1(x)} \right) \right| \\
&\leqslant \log (1 + \alpha e) \\
&= O(\alpha),
\end{aligned}
$$

where the at the second last step, the inequality follows from the fact that $d_k(\mathcal{D}_1, \mathcal{D}_1) \leqslant \alpha$, and $x < P^*_{F_1}$.

Further, $F_1(x) \geqslant F_2(x)$ for all $x \in \mathbb{R}_+$ implies that $h_1(x) \geqslant h_2(x)$ for all $x \in \mathbb{R}_+$. Therefore, $h_1(P^*_{F_1}) = \log(P^*_{F_1}) \geqslant h_2(P^*_{F_1})$. Therefore, we have $P^*_{F_2} \leqslant P^*_{F_1}$, and

$$
|h_1(P^*_{F_2}) - h_2(P^*_{F_2})| \leqslant \log (1 + \alpha e).
$$

Now define functions $s_1(x) = h_1(x) - \log(x)$, and $s_2(x) = h_2(x) - \log(x)$. Then by the definition of $P^*_{F_1}$, $P^*_{F_2}$ and Lemma A.1,

$$
\begin{aligned}
\min_{x \leqslant P^*_{F_1}} s_1(x) = s_1(P^*_{F_1}) &\leqslant s_1(P^*_{F_2}) \\
&\leqslant s_2(P^*_{F_2}) + \log (1 + \alpha e) \\
&= \min_{x \leqslant P^*_{F_2}} s_2(x) + \log (1 + \alpha e).
\end{aligned}
$$

Therefore, by the definitions of $s_1$ and $s_2$,

$$
\begin{aligned}
&\left| \min_{x \leqslant P^*_{F_1}} s_1(x) - \min_{x \leqslant P^*_{F_2}} s_2(x) \right| \leqslant \log (1 + \alpha e) \\
&\iff |\log(\mathrm{OPT}_{F_2}) - \log(\mathrm{OPT}_{F_1})| \leqslant \log (1 + \alpha e) \\
&\iff -\log (1 + \alpha e) \leqslant \log(\mathrm{OPT}_{F_2}) \leqslant \log (1 + \alpha e) \\
&\iff (1 + \alpha e)^{-1} \leqslant \mathrm{OPT}_{F_2} \leqslant 1 + \alpha e.
\end{aligned}
$$

The above directly implies:

$$
(1 + \alpha e)^{-1} \leqslant \frac{\mathrm{OPT}_{F_1}}{\mathrm{OPT}_{F_2}} \leqslant 1 + \alpha e.
$$

which completes the proof. ■

Now we are ready to prove Theorem 3.6 for the $n = 1$ case.

*Proof.* First, by construction, Algorithm 1 runs the Myerson optimal auction on an MHR distribution $\widehat{F}$, such that $\widehat{F} \geqslant \widehat{F}'(x)$ for all $x \in \mathbb{R}_+$, for any MHR distribution $F'(x)$ such that $d_k(F'(x), \tilde{F}(x)) \leqslant \alpha$. Also by assumption, $d_k(F^*(x), \tilde{F}(x)) \leqslant \alpha$. Therefore by triangle inequality, $d_k(F^*(x), \widehat{F}(x)) \leqslant d_k(F^*(x), \tilde{F}(x)) + d_k(\tilde{F}(x), \widehat{F}(x)) \leqslant 2\alpha$.

Denote $\alpha' = 2\alpha$. By Proposition B.5,

$$
(1 + \alpha' e)^{-1} \leqslant \frac{\mathrm{OPT}_{F_1}}{\mathrm{OPT}_{F_2}} \leqslant 1 + \alpha' e.
$$

Note that $(1 + \alpha' e)^{-1} = (1 + 2\alpha e)^{-1} = 1 - O(\alpha)$, which completes the proof. ■

# C   Proof of Optimality for the Upper Bounds

For these lower bounds we follow the idea of the lower bounds from Guo et al. [2019] adapted to the corrupted case that we consider in this paper. The lower bound constructions of Guo et al. [2019] are based on a family of distributions

$$\mathcal{H} = \{\mathbf{D} \mid \mathcal{D}_1 = \mathcal{D}^b, \mathcal{D}_i = \mathcal{D}^h \quad \text{or} \quad \mathcal{D}_i = \mathcal{D}^\ell \text{ for all } 2 \leqslant i \leqslant n\}.$$

Observe that this family is characterized by the triplet of distributions $\mathcal{D}^b$, $\mathcal{D}^h$, and $\mathcal{D}^\ell$ for which we ask for the following conditions.

a) $\mathcal{D}^b$ is a point mass at $v_0$.

b) The propability of $v \geqslant v_2$ is at most $1/n$ both when $v \sim \mathcal{D}^h$ and when $v \sim \mathcal{D}^\ell$.

c) The probability of $v_1 > v \geqslant v_2$ is at least $p$ both when $v \sim \mathcal{D}^h$ and when $v \sim \mathcal{D}^\ell$.

d) For any value $v$ such that $v_1 > v \geqslant v_2$, we have $\phi^\ell(v) + \Delta \leqslant v_0 \leqslant \phi^h(v) - \Delta$, where $\phi^\ell$ is the virtual value function of $\mathcal{D}^\ell$ and correspondingly for $\phi^h$.

e) For any value $v$ such that $v < v_2$, we have that $\phi^h(v), \phi^\ell(v) \leqslant v_0$.

f) For any value $v_1 > v \geqslant v_2$ we have that the ratio $\frac{d\mathcal{D}^h}{d\mathcal{D}^\ell}(v)$ is upper and lower bounded by a constant, where $\frac{d\mathcal{D}^h}{d\mathcal{D}^\ell}$ is the Radon–Nikodym derivative between $\mathcal{D}^h$ and $\mathcal{D}^\ell$.

g) $\mathcal{D}^h$ is regular.

h) The point $v_1$ is either $+\infty$ or is a point mass and an upper bound on the support in both $\mathcal{D}^\ell$ and $\mathcal{D}^h$.

Under these conditions and using the exact same proof as the Lemma 18 from Guo et al. [2019] we can show the following.

**Lemma C.1.** *Let $\mathcal{H}$ be a class of distributions that satisfies the conditions a) - h) and additionally satisfies the following.*

i) *We have that $d_k(\mathcal{D}^\ell, \mathcal{D}^h) \leqslant \alpha/n$.*

*Then any algorithm that is robust to a total corruption $\alpha$ in Kolmogorov distance across all bidders achieves revenue of at most*

$$\text{OPT}(\mathbf{D}) - \Omega(n \cdot p \cdot \Delta)$$

*for any distribution $\mathbf{D} \in \mathcal{H}$.*

## C.1   MHR Distributions – Proof of Theorem 3.7

Let $a = \ln(n) - \ln(1-\beta)$, $b = \ln(n)$, $v_0 = a - 1$, $v_1 = \ln(n) - 2 \cdot \ln(1-\beta)$, $v_2 = a$, $p = \beta \cdot (1-\beta)/n$, $\Delta = 1/2$. Then we define $\mathcal{D}^\ell$ and $\mathcal{D}^h$ according to their CDFs $F^\ell$ and $F^h$ which are the following:

$$F^\ell(v) = \begin{cases} 1 - \exp(-v) & v < v_1 \\ 0 & v \geqslant v_1 \end{cases},$$

$$F^h(v) = \begin{cases} 1 - \exp\left(-\frac{b}{a} \cdot v\right) & v < v_2 \\ 1 - \exp\left(-\frac{v_1 - b}{v_1 - a} \cdot (v - a) + b\right) & v_2 \leqslant v < v_1 \\ 0 & v \geqslant v_1 \end{cases}.$$

Observe also that for this choice of distributions it holds that

$$\phi^\ell(v) = \begin{cases} v - 1 & v < v_1 \\ v_1 & v \geqslant v_1 \end{cases},$$

$$\phi^h(v) = \begin{cases} v - \frac{a}{b} & v < v_2 \\ v - \frac{v_1 - a}{v_1 - b} & v_2 \leqslant v < v_1 \\ v_1 & v \geqslant v_1 \end{cases}.$$

Now the conditions a) - h) are easy to verify. For the condition i) we observe that the maximum difference between the two CDFs is at $v = v_2$ for which we have that $\left|F^\ell(v_2) - F^h(v_2)\right| \leqslant \beta/n$. Hence, Lemma C.1 implies that the maximum revenue achievable by any robust mechanism is

$$\text{OPT}(\mathbf{D}) - \Omega(n \cdot p \cdot \Delta) = \text{OPT}(\mathbf{D}) - \Omega(\beta).$$

Observe that since the maximum value of any bidder is at most $\ln(n)$ we have that the maximum revenue is

$$\left(1 - \frac{\beta}{\ln(n)}\right) \cdot \text{OPT}(\mathbf{D}).$$

If we write this expression with respect to the amount of corruption per bidder, then we have that the maximum possible revenue is

$$\left(1 - \frac{n \cdot \alpha}{\ln(n)}\right) \cdot \text{OPT}(\mathbf{D}).$$

Finally, we observe that all of $\mathcal{D}^b$, $\mathcal{D}^\ell$, and $\mathcal{D}^h$ are MHR and hence Theorem 3.7 follows.

### C.2 Regular Distributions – Proof of Theorem 3.9

For the case of regular distributions we will use the same distributions used by Guo et al. [2019] in their proof of their Theorem 2. In particular, let $v_0 = 3/2$, $v_1 = +\infty$, $v_2 = 1 + \frac{1}{\beta}$, $p = \frac{\beta}{n}$, and $\Delta = 1/2$. We define $\mathcal{D}^\ell$ and $\mathcal{D}^h$ through their CDFs as follows

$$F^\ell(v) = 1 - \frac{1}{n \cdot (v-1)},$$

$$F^h(v) = \begin{cases} 0 & v < 1 + \frac{1}{n} \\ 1 - \frac{1}{n \cdot (v-1)} & 1 + \frac{1}{n} \geqslant v < v_2 \\ 1 - \frac{1-\beta}{n \cdot (v-2)} & v \geqslant v_2 \end{cases}.$$

The fact that these distributions satisfy a) - h) can be found in Guo et al. [2019]. We will focus on proving i). It is not hard to see that the two CDFs appears when $v = \bar{v} = 1 + \frac{1}{\sqrt{1-\beta}}$. For this value we have

$$\left|F^\ell(\bar{v}) - F^h(\bar{v})\right| = \frac{1}{n}\left(2 - \beta - 2\sqrt{1-\beta}\right) \leqslant \frac{\beta^2}{n},$$

where the last inequality can be easily verifies for $\beta \leqslant 1$. Now setting $\alpha = \frac{\beta^2}{n}$, observing that $n \cdot p \cdot \Delta = \Omega(\beta)$, and observing that $\text{OPT}(\mathbf{D}) \leqslant O(1)$ we can apply Lemma C.1 and we get that the maximum possible revenue is

$$\left(1 - \Omega\left(\sqrt{n \cdot \alpha}\right)\right) \cdot \text{OPT}(\mathbf{D}).$$

Finally by observing that all of $\mathcal{D}^b$, $\mathcal{D}^\ell$, and $\mathcal{D}^h$ are regular Theorem 3.9 follows.

## D   Proofs of Sample Complexity Bounds

### D.1   Proof of Theorem 4.3, $n > 1$ Case

This follows easily from Theorem 3.8 and the DKW inequality Dvoretzky et al. [1956], Massart [1990] that states that the empirical CDF with $m$ samples is close to the population CDF with an error of at most

$$O\left(\sqrt{\frac{\log(1/\delta)}{m}}\right)$$

with probability at least $1 - \delta$. ∎

## D.2 Proof of Theorem 4.3, $n = 1$ Case

We present in this section a proof of Theorem 4.3 for the case with $n = 1$ and regular distributions. In this case, we show that Algorithm 2 achieves the optimal sample complexity, up to a poly-logarithmic factor.

First, by [Lemma 5, Guo et al. [2019]], we have that with probability at least $1 - \delta$, for any value $v \geqslant 0$, the quantiles of $\tilde{\mathcal{D}}$ and its empirical counterpart $E$ satisfy that:

$$|q^E(v) - q^{\tilde{\mathcal{D}}}(v)| \leqslant \sqrt{\frac{2q^{\tilde{\mathcal{D}}}(v)(1 - q^{\tilde{\mathcal{D}}(v)}) \ln(2m\delta^{-1})}{m}} + \frac{\ln(2m\delta^{-1})}{m}. \tag{20}$$

Further note that by construction, we have

$$q^E - q^{\widehat{E}} \leqslant \sqrt{\frac{2q^E(v)\left(1 - q^E(v)\right) \ln(2m\delta^{-1})}{m}} + \frac{4\ln(2m\delta^{-1})}{m} + \alpha.$$

Given that Algorithm 2 runs the Myerson optimal auction on $\tilde{E}$, which is a minimal regular distribution that dominates $\tilde{E}$. Further, $\widehat{E} \succeq D^*$ by construction, assuming Eq (20) holds. Therefore, we have $D^* \succeq \tilde{E}$ assuming Eq (20) holds. Applying Lemma 3.3 yields:

$$\text{Rev}(M_{\tilde{E}}, \mathcal{D}^*) \geqslant \text{Rev}(M_{\tilde{E}}, \tilde{E}) = \text{OPT}(\tilde{E}).$$

Therefore, the remaining task is to ensure that $m$ is sufficiently large such that

$$\text{OPT}(\tilde{E}) \geqslant (1 - \sqrt{\alpha})\text{OPT}(\mathcal{D}^*).$$

We will use a useful lemma below which connects the ratio of revenues that we are interested in with the value of link function at an optimal reserve price.

**Lemma D.1.** *Given two regular distributions $\mathcal{D}, \bar{\mathcal{D}}$ with CDFs $F, \bar{F}$, such that $\bar{F} \succeq F$ and $d_k(\mathcal{D}, \bar{\mathcal{D}}) \leqslant \beta$. Denote the optimal reserve price for $\bar{F}$ as $\bar{P}$, and the optimal expected revenue for $F, \bar{F}$ as $\text{OPT}_F, \text{OPT}_{\bar{F}}$. Then we have*

$$\frac{\text{OPT}_F}{\text{OPT}_{\bar{F}}} \geqslant 1 - \beta h_r(\bar{P})$$

*Proof.* Recall that $h_r(x) = \frac{1}{1 - F(x)}$, and $\bar{h}_r(x) = \frac{1}{1 - \bar{F}(x)}$. Then, $F(x) \geqslant \bar{F}(x)$ implies $h_r(x) \geqslant \bar{h}_r(x)$.

By definition, $d_k(\mathcal{D}, \bar{\mathcal{D}}) \leqslant \beta$ implies that $\max_x F(x) - \bar{F}(x) \leqslant \beta$. So we have:

$$h_r(x) - \bar{h}_r(x) = \frac{F(x) - \bar{F}(x)}{(1 - F(x))(1 - \bar{F}(x))} = (F(x) - \bar{F}(x))h_r(x)\bar{h}_r(x) \leqslant \beta h_r^2(x),$$

where the last inequality follows from the fact that $\max_x F(x) - \bar{F}(x) \leqslant \beta$, and $h_r(x) \geqslant \bar{h}_r(x)$. Thus, for all $x$,

$$\bar{h}_R(x) \geqslant h_r(x) - \beta h_r^2(x). \tag{21}$$

Note that the expected revenue, $R(x) = x(1 - F(x))$, at any $x$, equals to $\frac{x}{h_r(x)}$, which is the reciprocal of the slope for the linear function $g(a) = h_r(x) \cdot a$. Hence, the revenue is maximized when the slope for the linear function $g(a) = h_r(x) \cdot a$ is minimized.

Denote the corresponding optimal reserve prices for $F$ and $\bar{F}$ as $P$ and $\bar{P}$. Then at $\bar{P}$,

$$\bar{h}_r(\bar{P}) = \frac{1}{1 - \bar{F}(\bar{P})} = \frac{1}{\text{OPT}_{\bar{F}}} \cdot \bar{P}.$$

Denote $\text{Rev}(F, x)$ as the expected revenue with a reserve price at $x$ for a valuation distribution with CDF as $F$. Then,

$$\frac{\text{OPT}_F}{\text{OPT}_{\bar{F}}} \geqslant \frac{\text{Rev}(F, \bar{P})}{\text{OPT}_{\bar{F}}} = \frac{\bar{h}_r(\bar{P})}{h(\bar{P})} \geqslant \frac{h_r(\bar{P}) - \beta h_r^2(\bar{P})}{h_r(\bar{P})} = 1 - \beta h_r(\bar{P}),$$

where the first inequality follows directly from the definition of the optimal revenue, and the second inequality is from Eq (21). ∎

Now we will use Lemma D.1 to proceed. Denote the optimal reserve price for $\mathcal{D}^*$ as $P^*$. Denote the link function applied to $\tilde{E}$ and $\mathcal{D}^*$ as $\tilde{h}$, $h^*$, respectively. Then, we will discuss two cases for $\tilde{h}(P^*)$.

**Case 1:** $\tilde{h}(P^*) > \frac{1}{\sqrt{\alpha}}$**.** For this case, $\tilde{h}(P^*) > \frac{1}{\sqrt{\alpha}}$ implies that $q^{\tilde{E}}(P^*) < \sqrt{\alpha}$. Applying [Lemma 5, Guo et al. [2019]] and triangle inequalities, we have

$$|q^{\tilde{E}} - q^{\mathcal{D}^*}| \leqslant \sqrt{\frac{2q^{\tilde{E}}(v)\left(1 - q^{\tilde{E}}(v)\right)\ln(2m\delta^{-1})}{m} + \frac{4\ln(2m\delta^{-1})}{m}} + \alpha.$$

Given that $q^{\tilde{E}}(P^*) < \sqrt{\alpha}$, we have $q^{\tilde{E}}(1 - q^{\tilde{E}}) \leqslant q^{\tilde{E}} \leqslant \sqrt{\alpha}$. Therefore, it suffices to have

$$\sqrt{\frac{\sqrt{\alpha}}{m}} \leqslant C_1\alpha,$$

for some universal constant $C_1$ to ensure that $|q^{\tilde{E}} - q^{\mathcal{D}^*}| = O(\alpha)$, which implies $m \geqslant 1/\{C_1^2\alpha^{3/2}\}$ for some universal constant $C_1$.

**Case 2:** $\tilde{h}(P^*) \leqslant \frac{1}{\sqrt{\alpha}}$**.** For this case, $\tilde{h}(P^*) \leqslant \frac{1}{\sqrt{\alpha}}$ implies that $q^{\tilde{E}}(P^*) \geqslant \sqrt{\alpha}$.

By lemma D.1, we have that

$$\frac{\text{OPT}_{\tilde{E}}}{\text{OPT}_{\mathcal{D}^*}} \geqslant 1 - \beta\tilde{h}_r(P^*),$$

therefore it suffice to ensure that $1 - \beta\tilde{h}_r(P^*) \geqslant 1 - C_2\sqrt{\alpha}$ for some universal constant $C_2$, which implies that $\beta \leqslant q^{\tilde{E}}(P^*) \cdot C_2\sqrt{\alpha}$. Applying [Lemma 5, Guo et al. [2019]], it suffices to have that $\sqrt{\frac{q^{\tilde{E}}(P^*)}{m}} \leqslant \beta \leqslant q^{\tilde{E}}(P^*) \cdot C_2\sqrt{\alpha}$, which yields that $m > \frac{1}{C_2^2\alpha q^{\tilde{E}}}$. Lastly, applying the fact that we are in the case where $q^{\tilde{E}}(P^*) \geqslant \sqrt{\alpha}$ we get that is suffices to have $m > \frac{1}{C_2^2\alpha^{3/2}}$ for some universal constant $C_2$. This completes the proof.

∎

## D.3 Proof of Theorem 4.4

This follows easily from Theorem 3.6 and the DKW inequality Dvoretzky et al. [1956], Massart [1990] that states that the empirical CDF with $m$ samples is close to the population CDF with an error of at most

$$O\left(\sqrt{\frac{\log(1/\delta)}{m}}\right)$$

with probability at least $1 - \delta$.

∎

## D.4 Proof of Theorem 4.5

We omit the details of this proof since it follows from Theorem 2 and Appendix E of Guo et al. [2019] applied for the case $n = 1$. The reason is that if we could get a better bound in our corrupted case then this algorithm could be used to improve our sample complexity result in the non-corrupted case.