# OpenReview forum: "Robust Learning of Optimal Auctions"
_NeurIPS.cc/2021/Conference — NeurIPS 2021 Spotlight_

### Official Review · Reviewer_gj2u · 2021-07-14

**Rating:** 7
**Confidence:** 3

**Summary:**

The paper studies the problem of designing approximately revenue-optimal auctions using corrupted / inaccurate priors, where inaccuracy is measured by the Kolmogorov-Smirnov distance, i.e., the maximum difference in the two CDFs.  The authors consider the multi-bidder single-item setting, and focus on the cases of regular and MHR prior distributions.  Notably, they do not assume bounded support.  The main results are almost matching (multiplicative) upper and lower bounds on the maximum revenue achievable given inaccurate priors with bounded KS distances to the real ones.  The authors further apply their results to sample-based settings, and derive similar upper bounds with polynomial number of samples from the inaccurate priors.

**Limitations And Societal Impact:**

I don't have concerns.

**Main Review:**

Strengths

The paper is overall well written.  The problem of auction design using corrupted / inaccurate priors is important, and has numerous potential applications (including auction design from samples as discussed in the paper).  The authors provide almost matching upper and lower bounds for this problem for both regular and MHR value distributions, which demonstrate a curious distinction between the two classes of distributions.  The technical contributions regarding inaccurate priors appear nontrivial (e.g., the use of link functions appears insightful, unbounded support and multiplicative approximations are subtle but quite often technically challenging).


Weaknesses

While I'm overall positive about the paper, my main concern is whether the paper makes sufficient *conceptual* contribution to the problem (I think the technical contribution is well justified).  Quite some prior work explicitly or implicitly (primarily dealing with problems such as dynamic auctions, auction design from samples, etc.) studies auction design with inaccurate priors, and I think (and I could be wrong) that some previous results also rely on assumptions about the Kolmogorov-Smirnov distance, which emerge naturally in auction design from samples.  From the discussion in the paper, the impression I got is that the main conceptual difference between this paper and prior work is in unbounded distributions.  While this is certainly an issue worth addressing, I'd be more excited if the authors could provide further justifications regarding the importance of this particular issue, or point out other conceptual contributions of the paper (e.g., are these the first almost tight bounds?).


Detailed comments

Def 3.5: "d_k(F', F) \le F": F => \alpha

First paragraph in sec 3.2, "... can be found in Appendix G" (and several other places): do you mean Appendix B?

Thm 3.7: what happend to the 1/ln(n) factor in the proof in Appendx C.1?

Proof sketch of thm 3.8, first sentence, "F(x) is stochastically dominated by F(x) for all x": this can't be right...

Thm 4.3 (and 4.4): is this just because the empirical CDF uniformly converges to the population CDF at rate m^{-1/2}?

Line 313, "sample complexity upper bounds depend on sum of \alpha_i": doesn't it depend on max_i \alpha_i^2?  Also "depends" => "depend"

Line 315, discussion about lower bounds: it's a bit confusing to see this before seeing the lower bounds (thm 4.5)

**Time Spent Reviewing:**

3

---

> ### Author Response · Authors · 2021-08-10
> **Response for reviewer gj2u**
>
> Thank you very much for your thoughtful comments. Also thanks a lot for reading our paper carefully and catching some typos. We have corrected all the minor typos. Below are our responses to the main comments.
>
> $\textbf{(1)}$ *“While I'm overall positive about the paper, my main concern is whether the paper makes a sufficient conceptual contribution to the problem (I think the technical contribution is well justified)... While this (unbounded distributions) is certainly an issue worth addressing, I'd be more excited if the authors could provide further justifications regarding the importance of this particular issue”*
>
> Indeed there have been a lot of previous works that explore inaccurate priors in the context of auctions. However, most of these works assume some systematic inaccuracy of the priors, or had a different objective (e.g. welfare maximization) [1, 2] or did not provide finite sample results for unbounded distributions [3, 4]. To the best of our knowledge, our work constitutes the first analysis of the learnability of single-item optimal auctions with the hardest type of inaccuracy in the priors, i.e., adversarially corrupted priors for unbounded distributions. Of all of the learning problems in which robustness to adversarial inputs is necessary, auctions are surely one of the most important, given that actual, real-world economic value is being established.
>
> $\textbf{(2)}$ *“Proof sketch of theorem 3.8, first sentence: F(x) is stochastically dominated by F(x) for all x”*
>
> Thank you for catching this typo. The second $F(x)$ was missing a \bar notation. We have corrected it as: “$F(x)$ is stochastically dominated by \bar{F}(x) for all x”.
>
> $\textbf{(3)}$ *“Thm 3.7: what happened to the 1/ln(n) factor in the proof in Appendix C.1?”*
>
> In Theorem 3.7 we have stated the bound with $\tilde \Omega$. We derived the explicit bound with the ln(n) factor in Appendix C.1. We will make sure to include a note on this in the final version of the paper.
>
> $\textbf{(4)}$ *"in Thm 4.3 (and 4.4): is this just because the empirical CDF uniformly converges to the population CDF at rate m^{-1/2}?"*
>
> Indeed the multi-bidder results follow from the DKW inequality and the population results from Section 3 (Theorem 3.7 and 3.8). We want to mention though that the optimal result that we get for the single-bidder case in Theorem 4.3 is more involved and needs additional arguments on top of the DKW inequality (which is detailed in Appendix D.2).
>
> $\textbf{(5)}$ *“Line 313, ‘sample complexity upper bounds depend on the sum of \alpha_i’: doesn't it depend on max_i \alpha_i^2?”*
>
> Indeed this phrase is confusing. What we wanted to say is that for the finite sample result the revenue guarantee depends on $\sum_i \alpha_i$ whereas indeed the number of samples depends on $max_i \alpha_i^2$, as we can see in both Theorem 4.3 and 4.4. This dependence of the revenue guarantee on $\sum_i \alpha_i$ comes as a consequence from Theorem 3.7 and 3.8 for the population model. We will make sure to make this clarification clear in the final version of the paper.
>
>
> $\textbf{References for reviewer gj2u:}$
>
> [1] Computational Bundling for Auctions, Christian Kroer and Tuomas Sandholm, AAMAS 2013 https://www.cs.cmu.edu/~sandholm/computationalbundling.aamas15.fromACM.pdf
>
> [2] Posted pricing and prophet inequalities with inaccurate priors, P Dütting, T Kesselheim, EC 2019 http://paulduetting.com/pubs/inaccurate-priors_EC19.pdf
>
> [3] Yang Cai and Constantinos Daskalakis. Learning multi-item auctions with (or without) samples. FOCS 2017 http://ieee-focs.org/FOCS-2017-Papers/3464a516.pdf
>
> [4] Johannes Brustle, Yang Cai, and Constantinos Daskalakis. Multi-item mechanisms without item independence: Learnability via robustness. EC 2020. https://arxiv.org/abs/1911.02146

---

> > ### Comment · Reviewer_gj2u · 2021-09-01
> > **Thank you for your detailed response**
> >
> > The response is very helpful, and I don't have further questions.

---

### Official Review · Reviewer_fxPB · 2021-07-21

**Rating:** 8
**Confidence:** 4

**Summary:**

This paper studies the problem of revenue maximization in a single-item setting, with one or many bidders, when the value distribution of the bidders presented to the mechanism designer may be corrupted.

**Limitations And Societal Impact:**

Yes

**Main Review:**

This paper studies the problem of revenue maximization in a single-item setting, with one or many bidders, when the value distribution of the bidders presented to the mechanism designer may be corrupted. The mechanism designer either has exact knowledge of the adversarially corrputed distribution, or only sample access to it. In both these models, the paper develops mechanisms that obtain good revenue when compared to the original distribution. In particular, when we fix an original product distribution D*, and consider any corrupted product distribution \tilde{D}, s.t., for each i, \tilde{D_i} is within an \alpha_i Kolmogorov-Smirnov distance of D*, the algorithm can take any such \tilde{D} as input and output a mechanism that gets good revenue w.r.t. D*. The "goodness" is measured through an approximation factor, which is multiplicative, and is equal to (1-O(\sum_i \alpha_i)) for MHR distributions and is equal to (1-O(\sum_i \sqrt{\alpha_i})) for regular distributions.

The distributions are unbounded, and this consitutes the primary difference of this work from earlier works that assume bounded distributions and obtain additive approximations that are dependent on the upper bound H of the support. No result can be obtained for all distributions. So the previous results restrict through boundedness, and this result restricts through MHR and regularity. But the important part is that MHR/regularity restriction leads to development of new techniques in this work.

In particular, the paper develops the notion of link and inverse-link functions. For MHR, the link function is -ln(1-F(x)), and for regular it is 1/(1-F(x)). These link functions, apart from having nice properties like convex and non-decreasing, also provide an alternate characterization for the monopoly price and optimal revenue.  The optimal algorithm simply outputs the Myerson optimal auction for the minimal distribution in the Kolmogorov ball around the input distribution \tilde{D}, where the notion of minimality crucially uses the link function definition.

The paper also provides nearly matching upper and lower bounds on sample complexity in the sample model.

Overall, this is a pretty neat contribution to mechanism design literature, and is a good fit for NeurIPS.

Specific comments:

1) Page 4, line 2: it should be \mu and \nu. Currently it is written as \mu and \mu.
2) Definition 3.5: give --> given
3) Definition 3.5: The quantity after \leq should not be F I think; it should be \alpha? Also, the ball is currently defined for one bidder distributions. Extension to product distribution is clear, but would be nice to explicitly spell it out.
4) Definition 3.5 is written in terms of cdfs. This is understood, but good to explicitly mention this.
5) In the definition of \hat{h} in equation 1: under the max it says F is MHR/regular. What is F? Do you mean F-bar? Also, in general it is good to avoid the F-bar notation as it is sometimes used to refer to 1-F.
6) In the \hat{F} that is finally computed in Equation (1): how do we ensure that it is MHR/regular? If the F-bars that get used are MHR/regular, then the \hat{F} that comes out is also MHR/regular I guess? Please expand a bit on it.

**Time Spent Reviewing:**

4.5

---

> ### Author Response · Authors · 2021-08-10
> **Response for reviewer fxPB**
>
> Thank you very much for your thoughtful comments. Also thanks a lot for reading our paper carefully and pointing out these typos. We have corrected all the typos 1-4.  Below are some clarifications with respect to 5 and 6.
>
> #5:  *“In the definition of $\hat{h}$ in equation 1: under the max it says F is MHR/regular...Do you mean F-bar?”*
>
> Thanks for catching this typo. Indeed it should be F-bar under the max. We have also made a revision to avoid using the F-bar notation in this definition, as it suggested by the reviewer.
>
> #6: *“In the $\hat{F}$ that is finally computed in Equation (1): how do we ensure that it is MHR/regular? Please expand a bit on it.”*
>
> We show in Lemma 3.4 that, any MHR/regular distribution can be equivalently characterized by a corresponding convex and increasing link function. Therefore, in order to guarantee that minimal distribution $\hat{F}$ that is finally computed in Equation (1) is MHR/regular, we only need to find the point-wise largest link function h that is convex and increasing, within $B_{d_k, \alpha}(F)$. Figure 2 illustrates an example for the regular distribution case, where since the $\hat h$ function is convex and increasing, it corresponds to a regular distribution $\hat F$ by inverting the link function. Empirically with sample access, we show in Lemma 4.2 that a similar procedure can be computed efficiently. We will make sure to add these clarifications to the paper in the final version.

---

### Official Review · Reviewer_WsiF · 2021-08-02

**Rating:** 6
**Confidence:** 4

**Summary:**

This paper studies the problem of learning revenue-optimal single-item auction using bidder valuation samples with adversarial corruptions.

The paper first studies the case when there’s a given distribution. When the true distribution is alpha-close to the given distribution in Kolmogorov-Smirnov distance, the paper shows an algorithm to to get at least 1-O(alpha) of the optimal revenue for regular distributions and at least 1-O(sqrt(alpha)) of the optimal revenue for MHR distributions. The paper also provides negative results to show the algorithms are tight.

The paper also gives an algorithm of approximating optimal auction when only samples from the given distribution are provided.


**Limitations And Societal Impact:**

Limitations are discussed above.

**Main Review:**

Robust learning optimal auctions is a very important problem. In general the paper is well-written.

The paper’s results for a given distribution are pretty complete. The only comment is that it would be better if the lower bounds (Thm 3.7, 3.9) provided can be improved to prove for any vector of (alpha_1,...,alpha_n). And it might be better to make it clear that to what extent these lower bounds show the algorithm is optimal (the algorithm would look more optimal if a more instance-dependent lower bound can be shown).

I have some comments about the results for the sample access case.
1) The bounds are not tight and make the results less complete.
2) The paper chooses to study the case when the distribution is adversarially corrupted. I am wondering how much it can say about the case when some samples are adversarially corrupted.
3) In the sample access setting, for a given corrupted distribution, an algorithm should tradeoff between the revenue guarantee and the number of samples. The paper’s theorem only gives one point on this tradeoff curve. I am wondering what will be the complete picture of the tradeoff (for example, by relaxing distance guarantee, you can get more tradeoff points).
4) It might be good to compare the performance of the paper’s algorithm in the case with 0 corruptions to prior work to get a sense of how tight the result is.


**Time Spent Reviewing:**

2 hours

---

> ### Author Response · Authors · 2021-08-10
> **Response for reviewer WsiF**
>
> Thank you very much for your detailed and thoughtful comments, and for engaging with the essential aspects of our work! Below are our point-by-point responses to the main comments, and we will include these into the paper.
>
> $\underline{Comment\ about\ the\ lower\ bounds}$
>
> *“The only comment is that it would be better if the lower bounds provided can be improved..and it might be better to make it clear that to what extent these lower bounds show the algorithm is optimal”*
>
> We agree that this is a good point, and an instance-dependent lower bound is possible. We chose to present a simpler version because the generalization requires proving a generalized statement of Lemma 18 in [1] where instead of O(n p \Delta) in the lower bound we have O(\sum_{i=1}^n p_i \Delta_i), which significantly complicates our current proof. We will add a discussion about this in the paper as a remark after the corresponding theorems. If the reviewer believes it is important to have a full proof of this statement, we are willing to add the generalized proof in the final version, although most of it will necessarily go to the appendix.
>
> $\underline{Other\ questions\ and\ comments}$
>
> $\textbf{(1)}$ *“The sample complexity bounds are not tight and make the results less complete”*
>
> We would like to note that the sample complexity bound (Theorem 4.3) that we derived for the single-bidder case with regular distribution is indeed tight. For the multiple bidder case there is a discrepancy between the upper and the lower bound, although in this case we conjecture that the mechanism that we provide is still optimal. This is certainly a very interesting open problem, and we believe that it requires some new techniques and will drive future research. We will make sure to highlight this as an interesting open problem in the introduction section in the final version of the paper.
>
> $\textbf{(2)}$ *“The paper chooses to study the case when the distribution is adversarially corrupted. I am wondering how much it can say about the case when some samples are adversarially corrupted.”*
>
> We would like to clarify that our model of corruption (line 136-137) in combination with the mechanisms that we proposed (algorithm 2) can actually easily handle the case of adversarially corrupted samples. The reason is that all of our mechanisms are based on the estimation of the quantiles of the bidders’ valuation distributions and are robust to adversarial changes up to $\alpha$ of those quantiles. It is easy to see that an adversarial corruption of at most an $\alpha$ fraction of the samples can change the quantiles by at most $\alpha$ in value. Therefore our mechanisms will still have the same guarantees that we describe in our Theorems even when a bounded fraction of the samples are corrupted. We will make sure to add and highlight this discussion in the final version of our paper.
>
> $\textbf{(3)}$ *“For a given corrupted distribution, an algorithm should tradeoff between the revenue guarantee and the number of samples... I am wondering what will be the complete picture of the tradeoff (for example, by relaxing distance guarantee, you can get more tradeoff points)”*
>
> Thank you for bringing up this comment. Indeed, our algorithm can be used to find a complete tradeoff curve. In particular, if we have an $\alpha_i$ corruption to the distribution of the $i$-th bidder and $m$ samples from the joint distribution, then the corresponding revenue guarantee of Theorem 4.3 becomes $\left(1 - O\left( \sqrt{\sum_i \alpha_i + \frac{n}{\sqrt{m}}} \right) \right)$ and the revenue guarantee of Theorem 4.4 becomes $\left(1 - \tilde{O}\left(\sum_i \alpha_i + \frac{n}{\sqrt{m}} \right) \right)$. It is also immediate to see that as m goes to $\infty$, the guarantees approach those for the population model. We will make sure to add this full curve tradeoff in our final version.
>
> $\textbf{(4)}$ *“It might be good to compare the performance of the paper’s algorithm in the case with 0 corruptions to prior work to get a sense of how tight the result is.”*
>
> When we have access to $m$ samples then:
> For MHR distributions: both the best known algorithm achieves $\left(1 - \tilde{O}\left(\frac{\sqrt{n}}{\sqrt{m}} \right) \right)$ ([1], table 2) fraction of revenue and our robust algorithm achieves $\left(1 - \tilde{O}\left(\frac{n}{\sqrt{m}} \right) \right)$ a fraction of revenue of the revenue. So the two are the same for a constant number of bidders, i.e., the dependence on $m$ is the same.
> For regular distributions: in the case of 0 corruptions, our analysis gives a revenue guarantee of $\left(1 - O\left(\frac{\sqrt{n}}{m^{1/4}} \right) \right)$ whereas the optimal known from prior work without corruptions is  $\left(1 - \tilde{O}\left(\frac{\sqrt{n}}{m^{1/3}} \right) \right)$ ([1], table 2).
>
> So there is a small discrepancy between our results for 0-corruptions and the best known results for 0-corruptions, but we are solving a much more difficult problem because our algorithm is also robust to adversarial corruption. This is exactly the same observation as the one we discussed in the answer to question 1. of the reviewer above.  We will make sure to add this discussion to the final version.
>
> $\textbf{References for reviewer WsiF:}$
>
> [1] Chenghao Guo, Zhiyi Huang, and Xinzhi Zhang. Settling the sample complexity of single-parameter revenue maximization. In Proceedings of the 51st Annual ACM SIGACT Symposium on Theory of Computing, 2019. https://arxiv.org/abs/1904.04962

---

> > ### Author Response · Authors · 2021-08-23
> > **Has our response addressed your concerns?**
> >
> > Hello reviewer WsiF, we would be grateful if you can confirm whether our response has addressed your concerns, and let us know if any issues remain.

---

> > > ### Comment · Reviewer_WsiF · 2021-09-02
> > > **Concerns addressed**
> > >
> > > Thanks for the response! My concerns have been addressed.

---

### Decision · Program_Chairs · 2021-09-27

**Decision:**

Accept (Spotlight)

**Comment:**

The paper studies the revenue-optimal mechanism design when the bidder value distribution can be adversarially corrupted, and gives tight bounds on the achievable approximation and the sample complexity. This is a nice contribution and a good fit to NeurIPS.